# APRICOT: Active Preference Learning and Constraint-Aware Task Planning with LLMs

**Huaxiaoyue Wang[1], Nathaniel Chin[1], Gonzalo Gonzalez-Pumariega[1], Xiangwan Sun[1], Neha Sunkara[1], Maximus Adrian Pace[1], Jeannette Bohg[2], Sanjiban Choudhury[1]**
[1]Cornell University, [2]Stanford University

**Abstract:** Home robots performing personalized tasks must adeptly balance user preferences with environmental affordances. We focus on organization tasks within constrained spaces, such as arranging items into a refrigerator, where preferences for placement collide with physical limitations. The robot must infer user preferences based on a small set of demonstrations, which is easier for users to provide than extensively defining all their requirements. While recent works use Large Language Models (LLMs) to learn preferences from user demonstrations, they encounter two fundamental challenges. First, there is inherent ambiguity in interpreting user actions, as multiple preferences can often explain a single observed behavior. Second, not all user preferences are practically feasible due to geometric constraints in the environment. To address these challenges, we introduce APRICOT, a novel approach that merges LLM-based Bayesian active preference learning with constraint-aware task planning. APRICOT refines its generated preferences by actively querying the user and dynamically adapts its plan to respect environmental constraints. We evaluate APRICOT on a dataset of diverse organization tasks and demonstrate its effectiveness in real-world scenarios, showing significant improvements in both preference satisfaction and plan feasibility. The project website is at `https://portal-cornell.github.io/apricot/`

**Keywords:** Active Preference Learning, Task Planning, Large Language Models

## 1 Introduction

For robots to perform personalized household tasks, they need to balance a user's preference with constraints in the user's household. Concretely, organizational tasks, such as putting away grocery items into a refrigerator, require the robot to place items based on where the user prefers and avoid colliding with the fridge or knocking other items over.

Large Language Models (LLMs) provide an effective interface for users to communicate what they want via natural language [1, 2, 3], but carefully articulating the task and their specific preferences can also become tedious. Conversely, it is often more straightforward for users to provide demonstrations of the task, from which Vision-Language Models (VLMs) can extract relevant states and actions. Thus, recent works [4, 5, 6, 7] have studied using LLMs to infer user preferences from a small set of such demonstrations. However, current approaches face two challenges. First, multiple preferences can consistently explain user behaviors in demonstrations. Randomly choosing one preference to generate a plan will fail to solve unseen initial conditions. Second, environmental constraints can invalidate some placement locations that satisfy the preference. Naively converting a preference to placement locations can cause the robot to collide with objects in the environment.

***Our key insight is to close the loop on LLMs, enabling them to refine preferences and plans through efficient interactions with both the user and the environment.*** We combine the generative

---

Correspondence to: Huaxiaoyue Wang, yukiwang@cs.cornell.edu

8th Conference on Robot Learning (CoRL 2024), Munich, Germany.

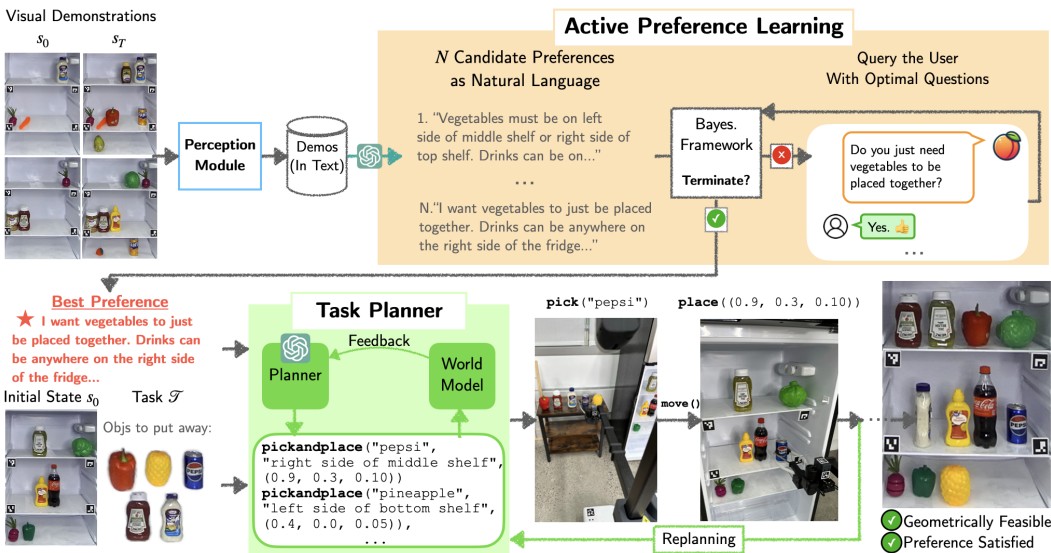

Figure 1: **Overview of** `APRICOT` that (1) converts user visual demonstrations into language-based demonstrations, (2) given demonstrations, determines the preference that best approximates the ground-truth user preference by minimally querying the user, (3) generates and refines a plan based on world models' feedback to satisfy preferences and respect constraints, (4) executes the plan in a real robot system.

question-asking capabilities of LLMs with Bayesian Active Learning [8] to explicitly capture uncertainty in user preferences and ask informative questions that collapse uncertainty. We also combine the generative planning capabilities of LLMs with feedback from the world model to iteratively improve plans, ensuring they respect environmental constraints while satisfying user preferences.

We present `APRICOT` (Active Preference Learning with Constraint-Aware Task Planner) that uses: (1) a VLM to convert visual demonstrations into language-based demonstrations; (2) an LLM-based Bayesian active preference learning module to ask the user a small number of questions and identify the preference that most closely approximates the user's preference; (3) a task planner that refines its robot task plan based on feedback from world models to satisfy preferences and respect environmental constraints; (4) a real robot system to execute the generated plan. Our contributions are:

- A new algorithm for LLM-based Bayesian active preference learning that can efficiently learn a preference from a small set of demonstrations and minimal online user-querying.

- A real robot system with a constraint-aware task planner that can generate and execute plans that satisfy user preferences and respect environmental constraints.

- Evaluations of (1) the active preference learning approach on a benchmark dataset of 50 different ground-truth preferences and 100 test cases, and (2) real robot experiments on 9 scenarios.

## 2    Related Works

**Active Preference Learning.**    Active preference learning focuses on estimating user preferences or reward functions by asking the user a set of queries. In robotics, classical approaches focus on synthesizing the optimal comparative queries (e.g., pairs of trajectories/features) [9, 10, 11, 12, 13, 14, 15, 16, 17, 18, 19]. A subgroup of works [20, 21, 22, 23] explores using teleoperated demonstrations to learn a prior over reward functions to expedite the process. In contrast, our work uses visual demonstrations. Instead of using a linear combination of features, we choose to represent preferences as natural language because language can capture complex preferences. Prior works [5, 4] have also studied using LLMs to infer user preferences based on demonstrations, but they lack an interactive mechanism to refine incorrect preferences based on user feedback.

Recent works explore incorporating LLMs into the active preference learning framework [24, 25, 26, 27]. [28] uses Bayesian optimal experimental design [29, 30] to select the optimal comparative pair of feature vectors and uses LLMs to translate the selected pair into natural language questions,

but feature vectors will fail to scale to our setting where placement preferences are combinatorial. Meanwhile, [31] uses LLMs to generate questions directly, but it relies solely on LLMs to pick effective questions to ask. Our approach bridges the two paradigms by proposing candidate questions to ask via LLMs and selecting the best questions that maximize information gain. Another tangential work is [32], which uses conformal prediction to quantify uncertainty, but it does not focus on active learning, and it requires a diverse calibration dataset, which is difficult to acquire for our setting.

**LLMs for Planning.** Work can be categorized as interactive task planners [33, 34, 3] that adapt the plan through user interactions, affordance planners [1, 35, 36] that generate feasible plans, and code planners [37, 4, 36, 2, 5] that generate plans as a sequence of function calls. Affordance planning is significant to our work since we must generate geometrically feasible plans to respect environmental constraints. Prior works like [38] verify its LLM plans with an external tool to solve structured planning domains. [35] verifies the feasibility of its LLM plan, which sequences robot skills, with learned Q-functions. Similar to these works, we use a world model to verify geometric feasibility. However, our work differs from both because the LLM planner incorporates and reflects on the world model's feedback for geometric feasibility to improve the plan over time [39, 40].

## 3 Problem Formulation

We frame the fridge organization task as an MDP. State $s \in S$ is the set of objects and their locations. Action $a \in A$ are high-level actions `pickandplace(obj_name, xyz_loc)`. Given a plan $\xi = \{s_0, a_0, \ldots, s_T, a_T\}$, the reward function $\mathcal{R}(\xi, \theta)$ evaluates whether object placements in $\xi$ satisfy the preference $\theta$. We represent these preferences as natural language, and $\theta^*$ is the (latent) ground-truth preference. We compute the reward $\mathcal{R}(\xi, \theta)$ through an LLM that outputs the percentage of object placements in $\xi$ that satisfy the preference $\theta$. We assume access to a world model, which computes geometric constraints $\mathcal{C}(\xi) = 0/1$ based on whether object placements cause collisions.

Given an initial condition $s_0$ (e.g., objects in the fridge), a task $\mathcal{T}$ (e.g., a set of objects to place), a preference prior $P(\theta)$, the goal is to find a feasible plan that maximizes the expected reward:

$$\arg \max_{\xi} \mathbb{E}_{\theta \sim P(\theta)} \mathcal{R}(\xi, \theta) \quad \text{s.t.} \quad \mathcal{C}(\xi) = 0. \tag{1}$$

To learn the prior $P(\theta)$, we leverage a small set of demonstrations $\mathcal{D}$ from the user similar to prior works [20, 21, 22, 23]. Given a plan $\xi_{\mathcal{D}} \in \mathcal{D}$, we model the prior as a Boltzmann distribution $P(\theta|\xi_{\mathcal{D}}) \propto \exp(\mathcal{R}(\xi_{\mathcal{D}}, \theta))$. Intuitively, preferences $\theta$ sampled from $P(\theta)$ are consistent with the demonstrations, i.e. rewards given these preferences are high: $\mathcal{R}(\xi_{\mathcal{D}}, \theta) = 1$. Hence, given demonstrations $\mathcal{D}$, we construct a preference prior $P(\theta|\mathcal{D}) = \prod_{\xi_{\mathcal{D}} \in \mathcal{D}} P(\theta|\xi_{\mathcal{D}})$. We make an important assumption that this prior has support over the ground truth preference $\theta^*$. [1]

One challenge is that demonstrations alone may not be sufficient to construct a good prior that helps solve (1). For example, multiple candidate preferences can explain the same demonstrations $\mathcal{D}$, but given a new initial condition unseen in the demonstrations, there may not exist a plan $\xi$ where $\mathcal{R}(\xi, \theta)$ is high for all preferences $\theta \sim P(\theta|\mathcal{D})$. Thus, we need an approach that can actively collapse uncertainty in $P(\theta)$ until a solution $\xi$ to (1), which has sufficiently high rewards given all plausible preferences, has been found. Another challenge is that if the LLM is directly used open-loop to generate a plan, it may not satisfy the constraints. We address both of these in the next section.

## 4 Approach

We introduce APRICOT, a three-stage approach, to tackle the problem of (1) learning user preferences from visual user demonstrations and a minimum number of queries (2) generating a plan that satisfies preferences and respects environmental constraints (3) executing the plan with a real-robot system.

### 4.1 Active Preference Learning

We present APRICOT's LLM-based Bayesian active preference learning module, which takes as input visual user demonstrations and converts them to language-based demonstrations $\mathcal{D}$ via a VLM

---

[1]We can relax this to prior has support over a preference $\hat{\theta}$ that is value equivalent to the ground truth preference $\theta^*$, i.e., the optimal plan $\xi^*$ according to $\hat{\theta}$ is also optimal according to $\theta^*$.

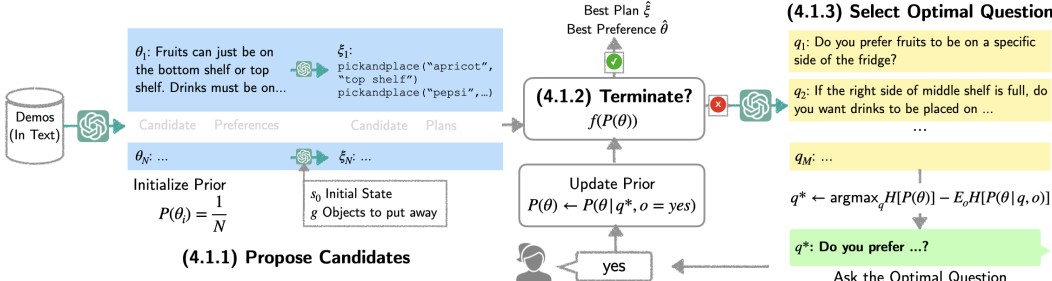

Figure 2: **LLM-Based Bayesian Active Preference Learning Approach.** Given a set of language-based demonstrations, APRICOT (1) proposes candidate preferences and corresponding candidate plans, (2) determines whether to terminate by evaluating whether the prior over candidate preferences $P(\theta)$ is sufficient, (3) select the optimal question that maximizes information gain before updating its prior based on user answers.

(Section 4.3). APRICOT outputs a preference that approximates the ground-truth user preference. Because user demonstrations alone cannot construct a sufficient preference prior (Section 3), the goal is to reduce uncertainty in this prior $P(\theta)$ by asking the user a small number of questions. This module contains three key components: (1) **Propose Candidate Preferences**, which generates candidates based on user demonstrations, (2) **Determine whether to Query**, which determines whether the module has sufficiently updated the prior, (3) **Select and Ask the Optimal Question**, which selects the optimal question to reduce uncertainty and updates the prior given user answers.

### 4.1.1 Propose Candidate Preferences

Directly maintaining the preference prior $P(\theta)$ as a continuous density function is challenging because preferences are represented as free-form natural language. Instead, APRICOT maintains a set of $N$ diverse candidate preferences $\theta \sim P(\theta|\mathcal{D})$ given a small set of user demonstrations $\mathcal{D}$. Concretely, we construct an LLM prompt that keeps sampling until getting preferences $\{\theta_i\}_{i=1}^{N}$ such that they are all consistent with the demonstrations, i.e. $\mathcal{R}(\xi_{\mathcal{D}}, \theta_i) = 1, \forall \xi_{\mathcal{D}} \in \mathcal{D}, i = 1, \ldots, N$. Having sampled $N$ preferences, we set the prior as $P(\theta_i) = \frac{1}{N}$. With this initial prior, given a new initial condition $s_0$ unseen in the demonstrations and a task $\mathcal{T}$, there is no guarantee that there exists a plan $\xi$ that satisfies all preferences, i.e., $\mathcal{R}(\xi, \theta_i) = 1$. This necessitates actively querying the user.

### 4.1.2 Determine Whether To Query

Given the current prior over candidate preferences $P(\theta)$, APRICOT determines whether it needs to ask more questions to reduce uncertainty or the current prior is sufficient to solve (1), i.e., there exists a plan $\xi$ that has sufficiently high rewards given all preferences in $\{\theta_i\}_{i=1}^{N}$.

To do so, we first construct a library $\Xi$ of candidate plans. Given an initial condition $s_0$ and a task $\mathcal{T}$, for each candidate $\theta_i$, we use the task planner (Section 4.2) to generate a plan $\xi_i$ that maximizes $\mathcal{R}(\xi_i, \theta_i)$. We accumulate $N$ plans in total $\Xi = \{\xi_i\}_{i=1}^{N}$ corresponding to the $N$ preferences $\{\theta_i\}_{i=1}^{N}$. We also define the *disadvantage function* $\text{DISADV}(\xi_j, \theta_i)$, which quantifies how bad a plan $\xi_j$ is compared to any other plan in the library $\Xi$ for a given preference $\theta_i$:

$$\text{DISADV}(\xi_j, \theta_i) = \max_{\xi \in \Xi} \mathcal{R}(\xi, \theta_i) - \mathcal{R}(\xi_j, \theta_i). \tag{2}$$

Then, the condition on whether to terminate querying $f(P(\theta))$ checks the following: does there exists a plan $\xi$ whose expected disadvantage across all candidate preferences is below a threshold $\epsilon$,

$$f(P(\theta)) = \exists \xi \in \Xi \quad \text{s.t} \quad \sum_{i=1}^{N} P(\theta_i)\text{DISADV}(\xi, \theta_i) \leq \epsilon. \tag{3}$$

Intuitively, APRICOT will terminate querying the user either (1) when a plan $\xi$ has low disadvantages across all candidate preferences or (2) when the probability of candidate preferences that give a high disadvantage to $\xi$ approaches 0. Once the terminating condition $f(P(\theta))$ is true, we get the best plan $\xi_i$ that satisfies (3) and, by proxy, the corresponding preference $\theta_i$ that generates the plan. Appendix 7.1.4 prove that APRICOT's performance is bounded given this terminating condition.

### 4.1.3  Select and Ask Optimal Query

To collapse uncertainty in $P(\theta)$, APRICOT asks the user a question $q$ and receives an answer $o \in \mathcal{O}$ that provides additional information about latent user preferences beyond the demonstrations. To efficiently query the user with informative questions, we choose questions $q^*$ that greedily maximizes information gain [29, 30]: $q^* \leftarrow \arg\max_q H[P(\theta)] - \mathbb{E}_o H[P(\theta|q, o)]$. Unlike the typical Bayesian approach that relies on a fixed corpus of questions, we use an LLM to generate a question set $\mathcal{Q}$.

For APRICOT, we assume that the robot will only ask yes-or-no questions, so user answers are defined as $o \in \{\text{yes}, \text{no}\}$. We use an LLM to generate $M$ questions for each preference pair $(\theta_i, \theta_j)$, which we accumulate to construct $\mathcal{Q}$. Empirically, the LLM often asks about specific aspects that are different between the pair instead of directly asking which preference is better (Fig. 4).

Given a candidate question set $\mathcal{Q}$, we can find the optimal question $q^* \in \mathcal{Q}$ by solving the equation:

$$q^* = \arg\max_{q \in \mathcal{Q}} \quad H[P(\theta)] - \sum_{o \in \{\text{yes}, \text{no}\}} \sum_{i=1}^{N} P(o|q, \theta_i) H[P(\theta_i|q, o)]. \tag{4}$$

To calculate the likelihood $P(o|q, \theta_i)$, we use the Bradley-Terry Model [41], where the score is the logit of a LLM $P^{\text{roleplay}}(.|q, \theta_i)$ that pretends to be a user with preference $\theta_i$ answering the question $q$. Thus, $P(o|q, \theta_i) = \sigma(P^{\text{roleplay}}(o = \text{yes}|q, \theta_i) - P^{\text{roleplay}}(o = \text{no}|q, \theta_i))$. The posterior $P(\theta_i|q, o) = \frac{P(o|q, \theta_i) P(\theta_i)}{P(o, q)}$ is the normalized product of the likelihood and prior. Once the user answers $o$ to the optimal question $q^*$, we update the prior with the posterior $P(\theta|q^*, o)$ and re-determine whether to terminate. Hyperparameters, computation cost, and LLM prompts are in Appendix 7.1.

## 4.2  Preference and Geometric Constraint Aware Task Planner

Given an initial condition $s_0$, a task $\mathcal{T}$, and a preference $\theta_i$, APRICOT's task planner must output a plan $\xi_i$ that maximizes the reward function $\mathcal{R}(\xi_i, \theta_i)$ and satisfies constraints $\mathcal{C}(\xi_i) = 0$ (Section 3).

**Semantic Plan.** We first prompt an LLM to output a semantic plan, which is a sequence of high-level semantic actions (e.g., `pickandplace("apricot", "top shelf")`).

**Geometric Plan Based on World Model.** Given a semantic plan, the task planner needs to ground it in geometrically feasible XYZ locations. We assume access to a world model, which, for the real-robot system, is represented as the 3D point cloud of the fridge and 3D bounding boxes for all objects. When calculating $\mathcal{C}(\xi_i)$, the world model collision checks an object's XYZ placement location with the point cloud. The task planner performs a beam search, where each node is an object's XYZ placement coordinate sampled based on the semantic plan's location. It selects the final $\xi_i$ based on the beam with the highest reward $\mathcal{R}(\xi_i, \theta_i)$ and lowest constraint violation $\mathcal{C}(\xi_i)$.

**Reflect and Refine Based on Feedback.** To refine the generated plan $\xi_i$, we construct a Reflexion [40] style prompt that takes feedback both from the reward $\mathcal{R}(\xi_i, \theta_i)$ and constraint violations $\mathcal{C}(\xi_i)$. We prompt the task planner to regenerate until it finds a feasible placement location for all objects or it exhausts its maximal refinement steps. See Appendix 7.2 for details.

## 4.3  Execution on a Real Mobile Manipulator

We introduce two additional components to implement APRICOT in a real-robotic system.

**Perception.** Given an image, the perception system detects objects and determines semantic locations for each object. We assume a list of all possible grocery items that can appear in a fridge, a static camera pointed at the fridge, and ArUco tags for shelf localization. We use an open-vocabulary object-detector Grounding-DINO [42] that uses the prompt "grocery item" to draw bounding boxes on all of the objects. After filtering with Non-Maximum Suppression, CLIP similarity score [43] determines the most likely label for a cropped object image. The semantic location (e.g. "top shelf", "middle shelf") for each object is determined by the pixel location of its bounding box center. The perception system is used to (1) parse a visual user demonstration into before and after states for a grounded, language-based demonstration and (2) track the current state of the fridge in real time.

**Execution Policy.** A high-level action `pickandplace(<obj>, <xyz_loc>)` is converted into a sequence of low-level skills. (1) For `pick(<obj>)` and `place(<xyz_loc>)`, we train a RL policy in a simple low-dimensional simulator that knows the robot's dynamic model. See details in Appendix 7.3. (2) For `move()`, we use a simple path planner. (3) To ensure the system is responsive to changes in the environment, we regenerate the plan $\xi_i$ after completing each high-level action.

# 5  Experiments

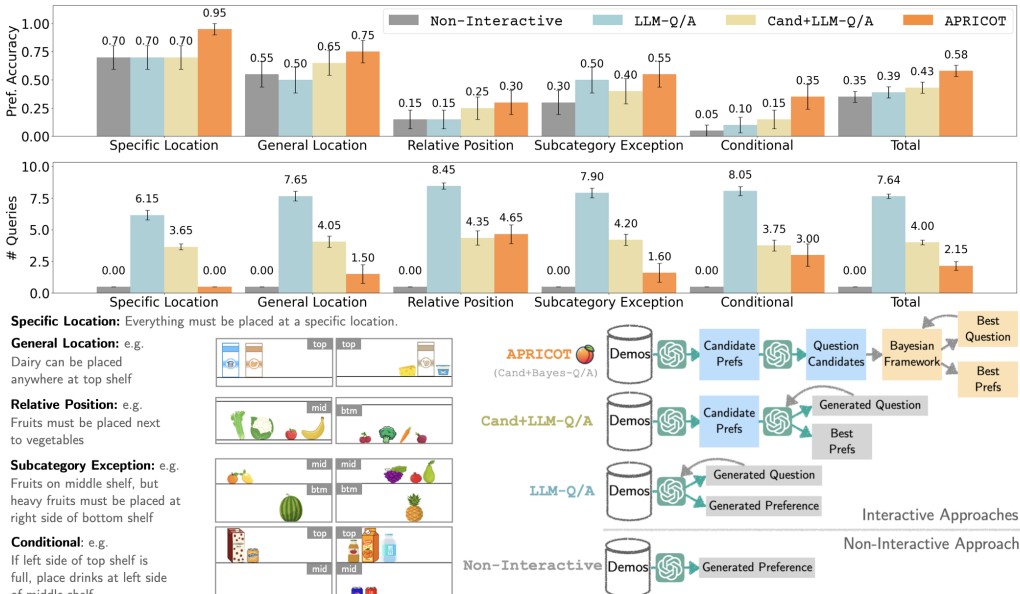

Figure 3: **Active Preference Learning Results on Benchmark Dataset.** `APRICOT` achieves the highest preference accuracy 58%, which is the percentage of outputted preferences that are equivalent to the ground-truth preference, while asking the user the smallest amount of questions (2.15 on average).

## 5.1  Active Preference Learning Experiment

**Benchmark Dataset.** To evaluate our active preference learning approach, we programmatically generate a dataset of 100 test cases containing a ground truth preference, 2 demonstrations, and a test scenario. The preferences always refer to object categories (e.g., fruits, vegetables), and they are split into 5 groups, each with 20 cases. Fig. 3 shows examples and Appendex 8.1.1 contains details.

**Setup.** We conduct the experiments in a 2D version of the fridge environment. The task planner generates a plan $\xi$ as a sequence of XY placement locations, and a 2D world model computes $\mathcal{C}(\xi)$ by collision checking object assets against the environment. To simulate a user, we use an LLM that can answer questions based on the ground-truth user preference. Details are in Appendix 8.1.2.

**Baselines.** We compare `APRICOT` against two categories of baselines: interactive and non-interactive ones. Fig. 3 shows visualizations of the differences between `APRICOT` and these baselines. Compared to `APRICOT`'s Bayesian framework, interactive baselines directly use LLMs to control the active preference learning process. `LLM-Q/A` relies on one LLM prompt to determine whether to stop asking questions, generate a question, or generate the preference when it terminates. `Cand+LLM-Q/A` splits the process into two steps, where it first generates candidate preferences similar to `APRICOT`. It then uses another LLM to determine whether to terminate, generate a question, or select the best preference from the candidates when it terminates. Meanwhile, `Non-Interactive` [5] uses LLMs to directly summarize the demonstrations into a preference.

**Metrics.** The **preference accuracy** represents the percentage of preferences determined by an approach that are equivalent to the ground-truth preference. Practically, based on Section 3, a preference $\theta_i$ is equivalent to the ground-truth preference $\theta^*$ if its corresponding plan $\xi_i$ is optimal according to $\theta^*$ such that $\mathcal{R}(\xi_i, \theta_i) = \mathcal{R}(\xi_i, \theta^*) = 1$. The **number of queries** quantifies the average number of questions an approach asks the user, so lower is better.

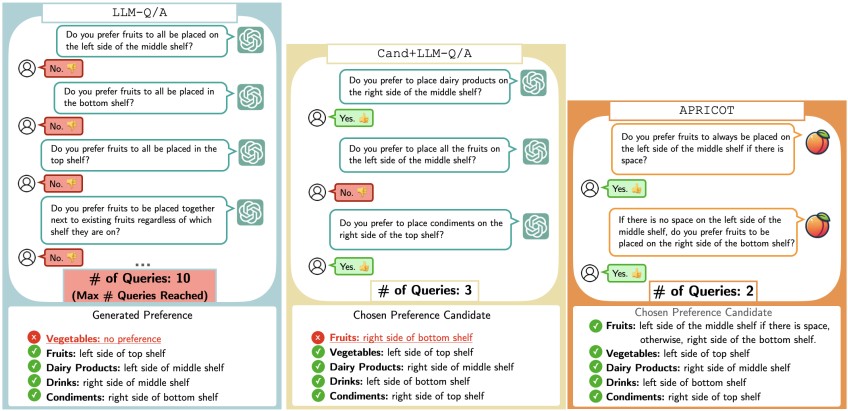

Figure 4: **Example Queries From Each Approach.** APRICOT correctly infers the ground-truth user preference with the least number of queries because it selects informative questions directly about the category with the most complex requirement. In contrast, LLM-Q/A exhausts the number of queries, while Cand+LLM-Q/A terminates early but infers the preference incorrectly. Preferences here are simplified as bullet points for readability.

**Q1: What is the quality of preferences determined by APRICOT compared to baselines?** In Fig. 3, APRICOT overall achieves the highest preference accuracy of $58.0\%$, while Non-Interactive performs the worst with only $35.0\%$ accuracy, LLM-Q/A with $39.0\%$ accuracy, and Cand+LLM-Q/A with $43.0\%$ accuracy. For easy test cases where demonstrations consistently show the same category at the same location, Non-Interactive performs equally well compared to the interactive baselines. However, as the demonstrations have more variability and the preferences become more complex, Non-Interactive's performance worsens. This trend suggests that interactive approaches, which learn additional information by querying the user, are useful in uncovering complex preferences. We analyze APRICOT's failure cases in depth in Appendix 8.1.3.

**Q2: How query-efficient is APRICOT compared to interactive baselines?** Fig. 3 shows that APRICOT consistently requires $71.9\%$ less queries compared to LLM-Q/A and $46.25\%$ less queries compared to Cand+LLM-Q/A. Fig 4 visualizes a qualitative example of each approach's behavior given the same demonstrations. LLM-Q/A struggles to perform well because its LLM prompt is burdened with simultaneously reasoning about the entire active preference learning process and about what preference to generate at the end. Thus, LLM-Q/A simply asks every possible preference for a category, causing it to exhaust the allocated number of queries. Meanwhile, Cand+LLM-Q/A asks fewer queries compared to LLM-Q/A, but its LLM incorrectly interprets the user's answer "no" and chooses the wrong candidate. In contrast, APRICOT's maximum entropy reduction objective (4) helps it select the most informative question to reduce uncertainty in the candidate preferences, while its terminating condition (3) helps it determine when it can terminate without excessive querying.

## 5.2 Real robot experiments

**Setup.** The task planner is tested on 9 different scenarios, categorized into 3 difficulty levels. Figure 5 shows examples of these scenarios. Each scenario is accompanied by 2 visual demonstrations, which we convert into language-based demonstrations and run APRICOT's active preference learning module to output a preference. We evaluate how well a task planning approach can generate a plan to satisfy this preference while respecting constraints. Details are in Appendix 8.2.1.

**Baselines.** The Feasibility-Only planner disregards user preferences and focuses only on geometric feasibility. It uses beam search to find collision-free placement coordinates for each object. The Preference-Only planner focuses on satisfying user preferences without considering geometric constraints, and its semantic plan is directly converted to XYZ coordinates.

**Metrics.** The **percentage of feasible plans** is the percentage of objects that have collision-free placement locations, which checks how well a planner is able to respect environmental constraints. Meanwhile, the **percentage of preference satisfied** is the percentage of objects that are placed at a location that satisfies the preference outputted by APRICOT's active preference learning module.

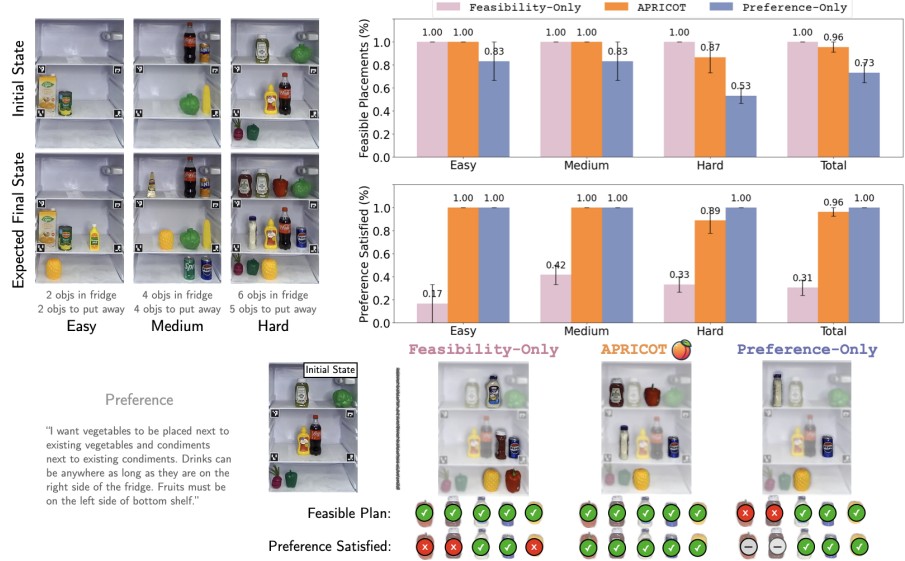

Figure 5: **Task Planner Results on Real-Robot Scenarios.** Evaluated on 9 scenarios with 3 difficulty levels. The qualitative example below APRICOT generating a plan that satisfies preferences and respect constraints

**Q3: How does APRICOT balance satisfying preference and respecting constraints?** In Fig. 5, even as the fridge space becomes more constrained, APRICOT is able to maintain a high percentage of feasible plan (96.0% for the Hard case) and preference satisfied (89.0%). Fig. 5 also shows a qualitative example of different planners' behavior. For this scenario, Feasibility-Only ignores the preference and just focuses on placing everything in the fridge. Preference-Only incorrectly assumes that the bottom shelf can fit both the red bell pepper and the pineapple, thereby causing the bell pepper to not receive a feasible placement location. In contrast, APRICOT initially makes the same incorrect assumption, but it is able to improve based on the world model's feedback and adjust its plan to place the red bell pepper on the right side of the top shelf. Appendix 8.2.2 shows detailed traces of APRICOT planner's improvements as it receives feedback.

**Q5: How does APRICOT regenerate plan to account for changes in the environment?** Fig. 6 shows that, because of its closed-loop planning, APRICOT can adapt its plan to environmental changes and continue satisfying user preferences. See Appendix 8.2.3 for other examples.

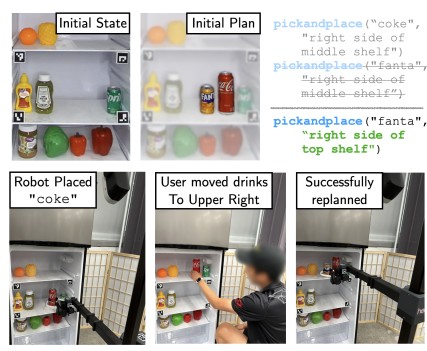

## 6 Discussion and Limitation

We explore the problem of solving a personalized organizational task with environmental constraints. We present APRICOT, a three-stage approach that learns the user's preference based on a small set of demonstrations and minimal online querying, generates a robot task plan that accounts for the learned preference and geometric constraints, and executes that plan on a real-robot system. Limitations include: (1) Active preference learning assumes one of the candidate preferences is equivalent to the ground-truth preference. Future work will explore dynamically expanding prior based on the question-answering and the initial demonstrations. (2) LLM task

Figure 6: APRICOT **Adapts to Changes in Environments.** The user's preference requires drinks to be placed together. After the user disrupts the scene and moves the drinks to the right side of the top shelf, APRICOT's closed-loop planner is able to adjust its plan and properly satisfy the user's preference in the newly changed environments.

planners cannot guarantee satisfying hard constraints even after our beam search and reflection steps. Future work will embed our LLM planner as fast heuristics within a complete search-based planner.

**Acknowledgments**

This work was supported in part by the National Science Foundation FRR (#2327973). Sanjiban Choudhury is supported in part by the Google Faculty Research Award and the OpenAI Superalignment Grant. We thank Marion Lepert and Jimmy Wu for discussing ideas for this project in its early stages.

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

# Appendix

## Table of Contents

## 7  Approach Details

### 7.1  Active Preference Learning

#### 7.1.1  Detailed Algorithm.

Algorithm 1 shows the detailed workflow of `APRICOT`'s active preference learning module. We note that, to expedite computation, we generate the candidate questions before entering the main active preference learning loop. For each candidate preference pair $(\theta_i, \theta_j)$, the question-generating LLM also receives the initial state $s_0$, the task $\mathcal{T}$, the candidate plans corresponding to the pair $\xi_i$, $\xi_j$, and each plan's rewards given one of the candidate preferences $(\mathcal{R}(\xi_i, \theta_i), \mathcal{R}(\xi_i, \theta_j))$ and $(\mathcal{R}(\xi_j, \theta_i), \mathcal{R}(\xi_j, \theta_j))$. During the main loop, once that question is asked, it is removed from the list of candidate questions.

#### 7.1.2  Hyperparameters.

We set the number of candidate preferences to $N = 5$ and the number of questions to generate for each preference pair to $M = 2$, so we generate 20 questions in total. For the terminating function, which determines when the preference prior $P(\theta)$ is sufficient, we set the terminating threshold to 0.07.

The components that require an LLM prompt are: generating candidate preferences, calculating the reward of a plan given a preference, generating candidate questions, estimating the likelihood of an answer given a preference and a question. All LLM calls use temperature of 0.7. We use `gpt-4` for "generating candidate preferences" because this task requires reasoning about the similarities and differences across multiple demonstrations and proposing a diverse set of preferences that are consistent with the demonstrations. We use `gpt-3-turbo` for "estimating the likelihood of answer" because this task calls the LLM the most number of times ($N \times M \times \binom{N}{2}$ times), and this task has a simple input-output space (inputs include a preference to roleplay and a yes-or-no question to answer; outputs include reasoning and yes/no answer). We use `gpt-4o` for the rest of the tasks. Prompts can be found in Appendix 9.1.

| Steps | # LLM calls | Avg. Tokens | Avg. Time (min) |
|---|---|---|---|
| Gen N preferences | 2 | 13000 | 2.5 |
| Gen N plans | 5 | 2800 | 0.25 |
| Calculate plans' rewards | 25 | 90000 | 2.5 |
| Propose candidate questions | 10 | 27000 | 1.16 |
| Calculate the likelihood | 100 | 200000 | 3.33 |
| One iteration of asking clarification questions | N/A | N/A | 0.0001 |

Table 1: Given $N = 5$, we report the number of LLM calls, the average number of tokens used, and the average wallclock time for each step in APRICOT's active preference learning module.

---

**Algorithm 1** APRICOT LLM-based Bayesian Active Preference Learning

---

**Input:** Demonstrations $\mathcal{D}$, Initial State $s_0$, Task $\mathcal{T}$, Terminating Function $f(P(\theta))$
**LLM Prompts:** LLM to generate candidate preferences $P^{\text{gen-pref}}$, LLM to generate plans $P^{\text{plan}}$, LLM to generate questions $P^{\text{gen-q}}$
**Output:** Best plan $\xi_i$, Corresponding preference of the best preference $\theta_i$
// Generate candidates
Generate candidate preferences $\{\theta_i\}_{i=1}^N$: $\theta \sim P^{\text{gen-pref}}(\theta|\mathcal{D})$
Initialize Prior $P(\theta_i) = \frac{1}{N}$
Generate candidate plans $\{\xi_i\}_{i=1}^N$: $\xi \sim P^{\text{plan}}(\xi|s_0, \mathcal{T}, \theta_i), i = 1, \ldots, N$
Construct $\mathcal{Q}$ by generating $M$ questions for all preference pairs: $q \sim P^{gen-q}(q|\theta_i, \theta_j)$
// While not exist a plan $\xi$ that has sufficiently high rewards given all preferences in $\{\theta_i\}_{i=1}^N$
**while** *not* $f(P(\theta))$ **do**
$\quad$ Find the best question: $q^* \leftarrow \arg\max_{q \in \mathcal{Q}} \quad H[P(\theta)] - \sum_o^{\{\text{yes,no}\}} \sum_i^N P(o|q, \theta_i) H[P(\theta_i|q, o)]$
$\quad$ Query the user with $q^*$ and gets answer $o_h$
$\quad$ Remove $q^*$ from $\mathcal{Q}$
$\quad$ Update prior: $P(\theta) = P(\theta|q = q^*, o = o_h)$
**Return** $\xi_i$ that satisfies the Terminating Function $f$, the corresponding $\theta_i$

---

### 7.1.3 Computation overhead.

For smooth user interaction, It is crucial that a system is fast when it actually asks user clarification questions.

APRICOT can complete an iteration of question-asking quickly (about 0.0001 min/iter) because it can find the optimal question by simply doing matrix calculations with the newly updated prior. All the necessary LLM calls, which naturally take longer, are completed before APRICOT starts asking questions. We also quantify the estimated token usage and wall-clock time for each step of the active preference learning algorithm in Table 1.

In addition, because APRICOT combines LLMs with active preference learning, it is able to scale well with increasing complexity (e.g. in preference types) because adding new preference types just requires adding a short description to the prompt. It can also scale well with an increasing number of user demonstrations. Because these demonstrations can become too long if they are simultaneously processed together, APRICOT can adapt the approach in [4], which summarizes each demonstration individually first before analyzing common patterns across all summaries.

### 7.1.4 Discussion About the Terminating Condition

We derive our termination condition as a principled approximation of the ideal termination condition, and provide additional analysis to quantify the performance gap due to the approximations.

**The Ideal Terminating Condition.**

Recall $\mathcal{R}(\xi, \theta)$ is the reward of a plan $\xi$ on whether it satisfies the preference $\theta$. We define a deterministic function $\mathcal{O}(q, \theta) = \arg\max_{o \in \{\text{yes}, \text{no}\}} P(o|q, \theta)$ that outputs the answer to an question $q$ given a preference $\theta$. Let $\theta^*$ be the ground-truth preference unknown to the system, $\mathcal{D}$ be a small set of demonstrated plans $\xi_{\mathcal{D}}$, and $\mathcal{H} = \{(q, o), \dots\}$ be a history of the questions asked so far and the corresponding answers from the user with $\theta^*$. We define the set of consistent preferences as

$$\Theta = \{\theta : \underbrace{(\forall \xi_{\mathcal{D}} \in \mathcal{D}, \mathcal{R}(\xi_{\mathcal{D}}, \theta) = 1)}_{\text{consistent with demos}} \wedge \underbrace{(\forall (q, o) \in \mathcal{H}, \mathcal{O}(q, \theta) = o)}_{\text{consistent with user answers}}\}. \tag{5}$$

The ideal terminating condition requires the module to keep asking questions until

$$\exists \xi \quad \text{s.t} \quad \forall \theta \in \Theta, \mathcal{R}(\xi, \theta) = 1. \tag{6}$$

However, it is computationally intractable to enumerate through all the possible preferences and compute optimal plans for each preference. We introduce a number of approximations.

**Approximation Assumptions.**

Approximating the aforementioned ideal terminating condition requires the following assumptions:

1. **When we sample a set of $N$ candidate preferences consistent with the demonstrations** $\hat{\Theta} = \{\theta_i : \forall \xi_{\mathcal{D}} \in \mathcal{D}, \mathcal{R}(\xi_{\mathcal{D}}, \theta_i) = 1\}_{i=1}^N$, **one of the sampled candidates $\theta$ is value-equivalent to the ground-truth preference $\theta^*$.** This candidate must be consistent with the user's answers, i.e., $\forall (q, o) \in \mathcal{H}, \mathcal{O}(q, \theta) = o$. Also,

$$\forall \xi, \|\mathcal{R}(\xi, \theta^*) - \mathcal{R}(\xi, \theta)\|_1 \leq \epsilon_{\text{realization}}. \tag{7}$$

2. **The task planner has a suboptimality of $\epsilon_{\textbf{planner}}$.** This planner's suboptimality lower bounds the suboptimality of the outputted plan. Given a preference $\theta$, let the optimal plan be $\xi^* = \arg\max_\xi \mathcal{R}(\xi, \theta)$. The performance difference between this optimal plan $\xi^*$ and the planner's plan $\hat{\xi}$ is:

$$\mathcal{R}(\xi^*, \theta) - \mathcal{R}(\hat{\xi}, \theta) \leq \epsilon_{\text{planner}}. \tag{8}$$

**Performance Bound Proof.**

**Theorem 7.1.** *Under assumptions (7) and (8), given a problem with the ground-truths $\theta^*$ and $\xi^*$,* APRICOT *outputs $\hat{\xi}$ when it follows (3) the terminating condition $f(P(\theta)) = \exists \xi \in \Xi \quad \text{s.t} \quad \sum_{i=1}^N P(\theta_i) \text{DISADV}(\xi, \theta_i) \leq \epsilon$ based on (2) the disadvantage function $\text{DISADV}(\xi_j, \theta_i) = \max_{\xi \in \Xi} \mathcal{R}(\xi, \theta_i) - \mathcal{R}(\xi_j, \theta_i)$. Its performance is bounded by*

$$\mathcal{R}(\xi^*, \theta^*) - \mathcal{R}(\hat{\xi}, \theta^*) \leq N\epsilon + 2\epsilon_{\text{realization}} + \epsilon_{\text{planner}}. \tag{9}$$

*Proof.* Let $\hat{\theta}$ be the candidate preference that is value-equivalent to $\theta^*$. Note that probabilities are non-negatives (i.e., $\forall \theta \in \hat{\Theta}, P(\theta) \geq 0$) and outputs of the disadvantage function are also non-negative (i.e., $\forall \xi, \forall \theta \in \hat{\Theta}, \text{DISADV}(\xi, \theta) \geq 0$). Based on the terminating condition (3), the worst case on $\hat{\theta}$ is:

$$P(\hat{\theta}) \text{DISADV}(\hat{\xi}, \hat{\theta}) \leq \epsilon \tag{10}$$

In other words, this scenario is when only one candidate $\hat{\theta}$ is value-equivalent to $\theta^*$, while all the remaining $N-1$ preferences $\hat{\Theta} \setminus \{\hat{\theta}\}$ agree on an incorrect plan $\hat{\xi}$ (i.e. $\forall \theta \in \hat{\Theta} \setminus \{\hat{\theta}\}, \mathcal{R}(\hat{\xi}, \theta) = 1$).

Expanding (10), we show:

$$\max_{\xi \in \Xi} \mathcal{R}(\xi, \hat{\theta}) - \mathcal{R}(\hat{\xi}, \hat{\theta}) < \frac{\epsilon}{P(\hat{\theta})} \qquad \text{via (2)} \tag{11}$$

$$\max_{\xi \in \Xi} \mathcal{R}(\xi, \theta^*) - \mathcal{R}(\hat{\xi}, \theta^*) \leq \frac{\epsilon}{P(\hat{\theta})} + 2\epsilon_{\text{realization}} \qquad \text{via (7)} \tag{12}$$

$$\mathcal{R}(\xi^*, \theta^*) - \mathcal{R}(\hat{\xi}, \theta^*) \leq \frac{\epsilon}{P(\hat{\theta})} + 2\epsilon_{\text{realization}} + \epsilon_{\text{planner}} \qquad \text{via (8)} \tag{13}$$

$$\mathcal{R}(\xi^*, \theta^*) - \mathcal{R}(\hat{\xi}, \theta^*) \leq N\epsilon + 2\epsilon_{\text{realization}} + \epsilon_{\text{planner}}. \tag{14}$$

The last inequality is via the fact that $P(\hat{\theta}) \geq \frac{1}{N}$. Because the initial prior is $\frac{1}{N}$ and the answers of $\hat{\theta}$ are consistent with the user's answers, its probability will never decrease. □

**Choose the terminating threshold $\epsilon$ in practice.** Similar to the proof, we consider the worst-case scenario before APRICOT starts asking any questions: $\hat{\theta}$ is the only candidate preference that is value-equivalent to the ground-truth preference $\theta^*$, and $\forall \theta \in \hat{\Theta} \setminus \{\hat{\theta}\}, \text{DISADV}(\hat{\xi}, \theta) = 0$ for an incorrect plan $\hat{\xi}$.

Ideally, APRICOT should not pick this incorrect plan, and it should ask clarification questions instead. To avoid satisfying the terminating condition (3), the following must be true

$$\epsilon \leq P(\hat{\theta})\text{DISADV}(\hat{\xi}, \hat{\theta}) \qquad\qquad\qquad\qquad\qquad \text{via (10)} \qquad (15)$$

$$\epsilon \leq \frac{1}{N}\text{DISADV}(\hat{\xi}, \hat{\theta}). \qquad\qquad \text{via } P(\hat{\theta}) \geq \frac{1}{N} \text{explained in the previous section} \qquad (16)$$

Thus, to set the terminating condition, we need to determine the minimum disadvantage that an incorrect plan can get given the correct candidate preference $\hat{\theta}$. We empirically find setting $x = 0.35$ has worked well, thereby resulting in threshold $\epsilon = 0.07$.

## 7.2 Task Planner

**Beam Search.** Recall that APRICOT's task planner first uses an LLM to output a semantic plan given a task $\mathcal{T}$ (a list of objects to put away) and an initial state $s_0$ (a dictionary of objects initially at each semantic location in the fridge). Then, it uses a beam search to find the XYZ placement location for each object by sequentially following the semantic plan.

The world model contains a mapping from a semantic location (e.g. "left side of top shelf") to a range of XYZ coordinates that corresponds to that region. With this semantic-location-to-geometric-range mapping, the beam search samples candidate XYZ placement location for each object based on its planned semantic place location. Specifically, it samples in XY space since objects can be placed anywhere length-wise and depth-wise on a shelf. Z location is based on the height of the shelf because we do not consider stacking objects atop one another. The best candidates are those where the placement is geometrically feasible (no fridge or object collisions). These candidates are maintained in the beam search nodes.

The beam search continues until all objects are placed into the fridge. An optimal plan is one where all objects acquire a geometrically-feasible, collision-free XYZ placement location. If the best plan in the search only places a subset of objects, the semantic plan is assumed to be geometrically infeasible and the LLM is reprompted with information about which objects lack a feasible XYZ placement location.

**Hyperparameters.** For semantic plan generation, we use `gpt-4-turbo` with a temperature of 0.7. For the beam search, we maintain 10 best candidates and sample 10 evenly-spaced points with the location regions. For feedback retries, we allow up to 4 attempts. Prompts can be found in Appendix 9.2.

## 7.3 Real Robot System

**RL Policy Training.** The policy `pick(<obj>)` and `place(<loc>)` can both be abstracted as the problem of given a goal XYZ position and the current robot joint state, output a sequence of actions that allows the gripper to reach the goal straight on. We simulate the robot's kinematics and include the $L_1$ norm between the goal and current positions as the observation space. We define the robot's teleoperation commands as the discrete action space. To guide the agent to learn a suitable grasp policy, we implemented a four-phase reward function:

1. Align the Gripper horizontally: The goal object should align in between the robot's gripper (X coordinate).
2. Align the Gripper height: The gripper should align with the goal height (Z coordinate).
3. Extend the gripper: The gripper should extend to pick/place the object (Y coordinate).
4. Goal Achievement: A reward of +1 is given if the robot successfully reaches the goal.

$$r = \begin{cases} -\Delta X + \max(\Delta Y) + \max(\Delta Z), & \Delta X \geq \epsilon_x \\ -\Delta Z - \Delta X + \max(\Delta Y), & \Delta Z \geq \epsilon_z, \Delta x \geq \epsilon_x \\ -\Delta Y - \Delta Z - \Delta X, & \Delta Y \geq \epsilon_y, \Delta Z \geq \epsilon_z, \Delta x \geq \epsilon_x \\ 1, & \Delta Y < \epsilon_y, \Delta Z < \epsilon_z, \Delta x < \epsilon_x \end{cases} \quad (17)$$

With this reward function, we train a Proximal Policy Optimization agent using the implementation from [44], to predict actions that will lead to a suitable grasp on the goal position.

# 8 Experiments Details

## 8.1 Active Preference Learning Experiments Details

### 8.1.1 Benchmark Dataset

The dataset contains 100 test cases, each containing a ground-truth preference, two demonstrations with before and after state of user putting objects into fridge, one scenario to solve. The ground-truth preference is created by (1) randomly generating the special preferences for 1 or 2 object categories and (2) randomly assigning simpler preferences for the remaining object categories. For the demonstrations and the test scenario, we randomly sample objects and specific placement coordinates that satisfy the generated ground-truth preference. Each demonstration has 3 objects initially in the fridge and 4 to place into the fridge. Meanwhile, each test scenario has 4 objects initially in the fridge and 6 to place into the fridge. The objects in the fridge can be categorized into 5 categories: fruits, vegetables, condiments, dairy products, juice and soft drinks. The test cases are also grouped into 5 categories, where each categories contain 20 test cases: specific location, general location, relative position, subcategory exceptions, and conditionals. Fig 3 shows examples of each category.

The test cases "specific location" require each category to be placed at a specific location in the fridge (e.g. "fruit on the left side of top shelf"), while the rest of the categories have special requirements on a subset of the categories and only specific location requirements on the remaining categories. For the 20 "general location" test cases, half of them require only one category to be at a general location (e.g. "fruits on the left side of the fridge"), and the other half require two categories to be at a general location (e.g. "fruits on the left side of the fridge, and vegetables on the right side of the fridge). In addition, for the 20 "relative position" test cases, half of them require one category to be together with no specific location (e.g. "fruits should be placed together regardless of what shelf they are on"), and the other half require two categories to be together (e.g. "fruits and vegetables should be placed together regardless of what shelf they are on.")

### 8.1.2 Setup

The prompts for the baseline approaches are in Appendix 9.3. We run each approach on a test case once. We make `APRICOT` generate $N - 1 = 4$ candidate preferences and add the preference generated by `Non-Interactive` as the fifth candidate preference. We also make `Cand+LLM-Q/A` have the same set of candidate preferences and candidate plans and `APRICOT` for a given test case.

### 8.1.3 Failure Modes of `APRICOT`

We find that `APRICOT` fails mainly because the LLMs hallucinate and make mistakes when they are called or used for evaluation.

After analyzing all the instances where `APRICOT` fails, we identify the following failure modes (with the most common failure modes sorted at the top):

1. **Incorrect question answering:** The LLMs can answer a question incorrectly due to hallucinating or interpreting the question incorrectly. This can be further categorized as:
   - **Mistakes during likelihood estimation** which is used to select the optimal questions.

| Incorrect QA | | No GT Pref | Mismatched | Incorrect reward | Demo |
|---|---|---|---|---|---|
| During likelihood est | During evaluation | | | | |
| 14.3% | 21.4% | 26.2% | 19.0% | 11.9% | 7.20% |

Table 2: Among the 42/100 test cases that `APRICOT` fails to solve, each column shows the percentage of failures that a caused by a specific failure mode.

- **Mistakes during evaluation** when the LLM pretends to be the user with the ground-truth preference and interacts with `APRICOT`.

2. **Ground-truth preference is not one of the candidate preferences:** Sometimes, the LLMs do not generate a candidate preference that fully captures the same requirements as the ground-truth preference.

3. **Mismatched understanding of object categories:** For example, some demonstrations use "tomatoes" as examples for vegetable placements because we treated tomatoes as vegetables, but the LLM classifies "tomatoes" as a fruit.

4. **Incorrect reward calculation:** The LLMs can make mistakes when calculating the reward of a plan and determining the percentage of object placements that satisfy a preference.

5. **Hallucinations when interpreting demonstrations:** The LLMs occasionally ignore some objects that appeared in the demonstrations.

Among the 42/100 test cases that `APRICOT` fails to solve, Table 2 reports the percentage of failures that are caused by each failure mode.

The most common failure mode, "Incorrect question answering" (causing 35.7% of the failures in total), explains why `APRICOT`'s performance is not higher. The LLMs that approximate the model of the human are imperfect. The subcategory "Mistakes during evaluation" causes 21.4% of the failures. When evaluation uses the LLMs to mimic the actual user, the LLMs can answer `APRICOT`'s question incorrectly, thereby causing `APRICOT` to also update its prior incorrectly.

For future work, we plan to improve the LLMs by (1) Switching to more powerful models: We currently use `gpt-3.5-turbo` to answer the questions, but we can switch to more powerful models such as `gpt-4o` or `Llama3` [45] to boost performances. (2) Aligning the LLMs' understanding of object categories with the user's: In the prompt, we can include information about how the user typically categorizes the objects that have ambiguous categories (e.g. "tomato," "bell pepper"). (3) Adding self-consistency verification: A verification module (which can be another LLM) can verify the output of the model [46, 47, 48].

### 8.1.4 Tradeoff between query efficiency and preference accuracy

We hypothesize that, ideally, as the number of queries increases, preference accuracy should also increase and eventually plateau.

However, because the most common failure mode is that the LLMs make mistakes when answering questions (see Section 8.1.3), we find that empirically asking too little and too many questions would both harm the performance. Specifically, we test on terminating threshold ranging from 0.09 to 0.03 and plot the number of queries v.s. preference accuracy in Figure 7. For the test cases under the more complex preference types ("subcategory exception" and "condition"), asking too many questions actually lowers the preference accuracy.

### 8.2 Task Planner Real Robot Experiments

### 8.2.1 Setup

Similar to the benchmark dataset for active preference learning experiments, the 9 real-world test cases (a ground-truth preference, two demonstrations, a test scenario) are randomly generated while

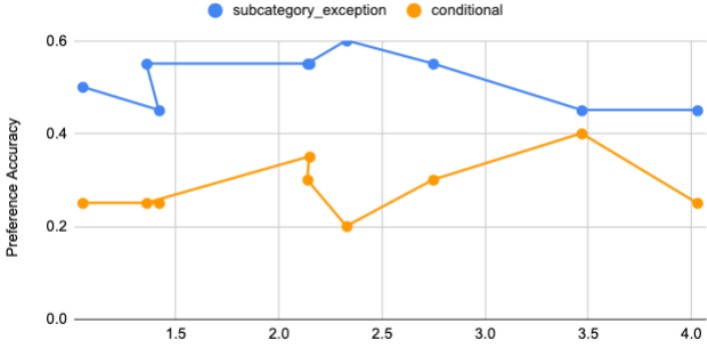

Figure 7: **The number of queries v.s. preference accuracy.** We test APRICOT on a terminating threshold ranging from 0.09 to 0.03. Contrary to the hypothesis, a large number of questions actually lowers the preference accuracy.

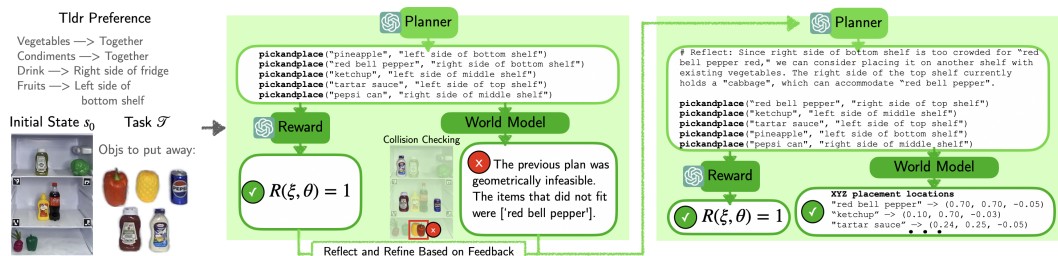

Figure 8: Example of APRICOT's task planner reflecting and refining its plan based on feedback.

following dataset generation rules to create scenarios where the geometric constraints are in tension with the preference. Each test case's ground-truth preferences are generated by first assigning some special preference for 1 to 3 object categories. For the demonstrations and the initial condition to test on, we randomly sample objects from a list of objects that we have, and we design each object's placement coordinates to be close to each other to create a cluttered environment. We show the visual demonstration and the ground-truth preference in Fig. 9.

For all real robot experiments, we use a 6-DoF Stretch Robot RE1 [49] as the mobile manipulator. A static RBG-D camera (ZED-2i [50]) is mounted to have a direct view of the fridge. A table is placed to the left of the fridge, on which objects that need to be put away by the robots are placed in a row with no occlusions. Before the experiments, we have mapped the environment and determined the location of the table and the fridge in the map. Images from the static camera are used by the perception module to track the current state of the world for the task planner and to convert user demonstrations into text-based demonstrations. The point cloud from the static camera is used by the world model to collision-check potential XYZ placement location. Meanwhile, the Stretch Robot's onboard RGB-D camera is used by the pick policy to identify the grasp position of an object.

We report the performance of APRICOT's active preference learning module in Table. 3.

### 8.2.2 APRICOT **Plan Refinement Example**

Fig. 8 shows an example of APRICOT's task planner refining its plan based on feedback. When the LLM generates the semantic plan, because it does not have knowledge about the environmental constraints of the fridge, it decides to place both the pineapple and the red bell pepper on the left side of the bottom shelf, which would satisfy the user's preference. However, only one object can be placed on the left side of the bottom shelf. Given this plan, although the reward function determines that the semantic plan maximally satisfies the preference $\mathcal{R}(\xi, \theta) = 1$, the world model determines

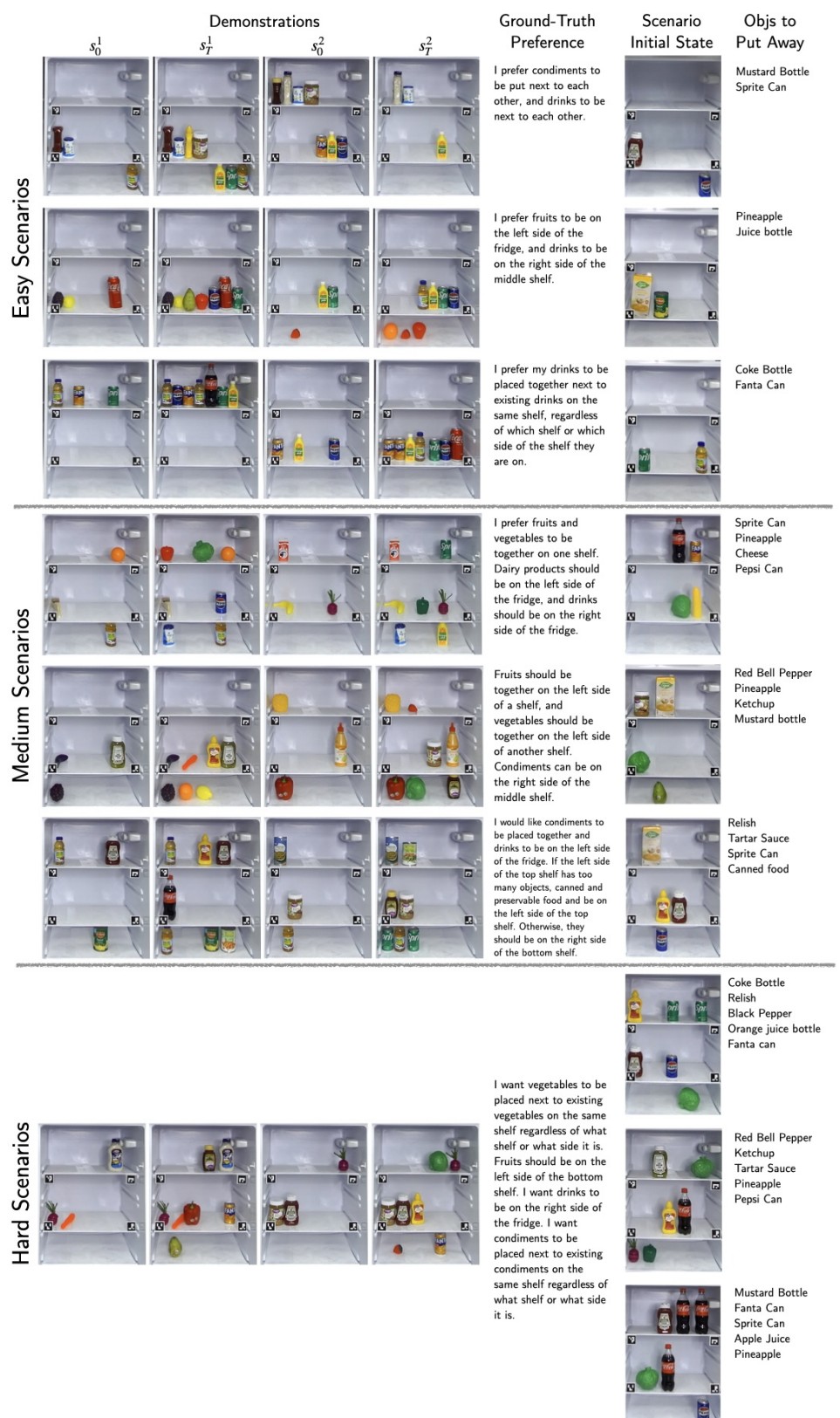

Figure 9: Real world demonstrations and scenarios.

|        | Scenario | Preference Accuracy | Number of Queries |
|--------|----------|---------------------|-------------------|
| Easy   | 1        | 1.00                | 0.00              |
|        | 2        | 0.00                | 5.00              |
|        | 3        | 1.00                | 0.00              |
| Medium | 1        | 0.00                | 0.00              |
|        | 2        | 1.00                | 1.00              |
|        | 3        | 0.00                | 0.00              |
| Hard   | 1        | 1.00                | 0.00              |
|        | 2        | 1.00                | 2.00              |
|        | 3        | 1.00                | 1.00              |
|        | Overall  | 0.56                | 1.00              |

Table 3: `APRICOT` active preference learning module's performance on real-world scenarios. Note that the difficulty level is based on how difficult the task is for the task planner (i.e. the number of objects initially in the fridge and the number of objects to put away.)

that red bell pepper will have a collision when being placed on the left side of the bottom shelf next to the pineapple. This feedback gets included in the input for the LLM, so the LLM is able to reflect on why its current plan is geometrically infeasible and determine an alternative semantic placement location for the red bell pepper ("right side of the top shelf"). The new plan satisfies the preference by maximizing the reward and is geometrically feasible by converting the semantic plans into XYZ placement locations for all the objects.

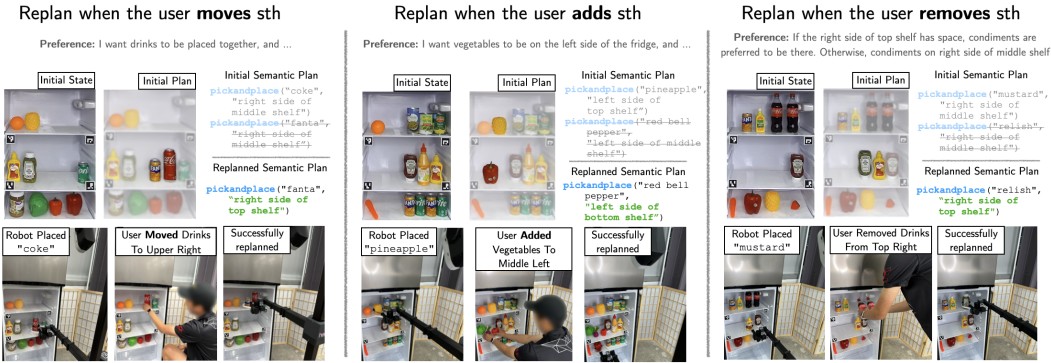

Figure 10: Examples of how `APRICOT`'s task planner can replan and handle different ways a user could affect and change the environment.

### 8.2.3  `APRICOT` **Replanning Example**

Fig. 10 show all 3 examples of how `APRICOT`'s task planner is able to replan and handle changes in the environment.

**Replanning after additions.** The user's preference requires vegetables to be placed on the left side of the fridge. The planner initially plans to place the bell pepper on the left side of the middle shelf. However, the user disrupts the scene and adds various vegetables there. `APRICOT`'s closed-loop planner adjusts to its plan by placing the bell pepper on the left side of the bottom shelf which still has space.

**Replanning after removals.** The user's preference requires placing condiments on the right of the top shelf if that space is empty. If that location is full, condiments can be placed on the right of the middle shelf. The planner initially plans to place the mustard and relish on the right side of the middle shelf since the right side of the top shelf is filled with coke bottles. However, the user

disrupts the scene and removes the bottles. `APRICOT`'s closed-loop planner adjusts to this by placing the relish in the freed space to best satisfy the user's preference.

### 8.2.4 Perception System Performance

To quantify the performance of our perception module, we evaluate it on the 9 real-world scenarios used in the task planner real-robot experiments.

**Baselines.** Recall that `APRICOT`'s perception module (`GD+CLIP`) uses Grounding-Dino [42] to detect the objects in the fridge and CLIP [43] to label the detected object name with an object name. Compared to this approach, we define the baseline `GD-Only`, which runs Grounding-DINO, given each object label from the list of all possible grocery items.

**Metrics.** We define two metrics. The **percentage of correctly identified object** calculates the average ratio between the detected objects that have the correct object label over the expected number of objects. The **number of incorrectly detected objects** calculates the average number of incorrect labels, so lower is better.

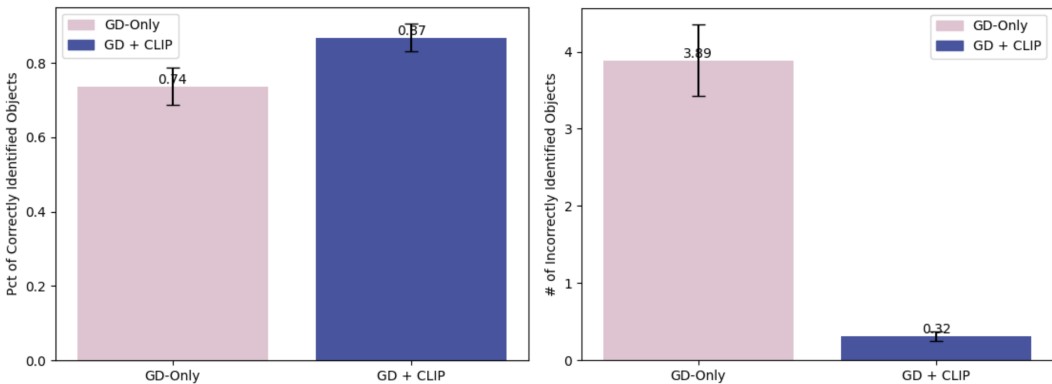

Figure 11: Result of perception module on real-robot scenarios. We also report the standard error in the bar plot.

**Results.** In Fig. 11, we observe that, for objects that are actually in the fridge, `GD+CLIP` is able to assign more correct labels ($87.0\%$) compared to `GD-Only` ($74.0\%$). In addition, `GD-Only` detects more objects incorrectly and has much more false positives compared to `GD+CLIP`. Because `GD-Only` calls Grounding-DINO for each possible object label, it repeatedly labels the same region in the image with different object labels. This behavior is undesirable because `APRICOT`'s task planner relies on the perception system to exactly detect the right number of objects in the fridge and assign each detected object with the correct label. `GD-Only` would cause the task planner to hallucinate what objects are currently in the fridge.

## 9 Prompts

### 9.1 Active Preference Learning Prompts

The prompts for `APRICOT`'s active preference learning modules are the following:

- Generate Candidate Preferences Given Demonstrations (Sec. 9.1.1)
- Generate a Plan Given a Preference, an Initial Condition, and a Task (Sec. 9.2)
- Calculate the Reward Given a Preference and a Plan (Sec. 9.1.2)
- Generate Candidate Preferences Given a Pair of Preferences (Sec. 9.1.3)
- Answer a Question Given a Preference (Sec. 9.1.4)

### 9.1.1 Generate Candidate Preferences

The LLM first analyzes the demonstrations in detail before proposing candidate preferences that can explain user behaviors in the demonstrations. We use `gpt-4` for these prompts because this is a complex task.

```
--------- System Message --------
You are an assistant who sees someone demonstrating how the fridge is
↪ organized and analyzes patterns in the demonstrations so that you can
↪ infer the potential preferences that the user might have.
--------- Instruction --------
 # Input
 You are given 2 demonstrations that show the before and after when a set
 ↪ of objects gets put into the fridge. For each demonstration:
 - "Objects that got put away" describes the objects that the user will
 ↪ demonstrate how they would like to put in the fridge.
 - "Initial state of the fridge" describes the objects that are initially
 ↪ in the fridge before the user starts the demonstration.
 - "Final state of the fridge" describes what the fridge looks like after
 ↪ the demonstration. All the objects in "Objects that got put away"
 ↪ should be in the fridge now.

 # Goal
 Your goal is to fill out the reasoning worksheet below that analyze the
 ↪ demonstrations and help you infer the preferences that the user could
 ↪ have.

 The demonstrations are complementary, and they are shown by the user with
 ↪  the same preference. Thus, even though they are different, their
 ↪ difference is due to the difference in the initial state of the fridge
 ↪  and the objects that got put away. They are not contradictory, and
 ↪ there exist preferences that can explain why the 2 demonstrations are
 ↪ the way they are.

 # Instructions and useful information
 ## Specific locations in the fridge
 The fridge can be segmented into 3 shelves (top shelf, middle shelf,
 ↪ bottom shelf); each shelf has 2 sides (left side, right side). The
 ↪ fridge has 6 specific locations:
 - left side of the top shelf, right side of the top shelf
 - left side of the middle shelf, right side of the middle shelf
 - left side of the bottom shelf, right side of the bottom shelf

 ## General locations in the fridge
 A general location contains multiple specific locations. There are 5
 ↪ general locations:
   - "left side of fridge" contains: "left side of top shelf", "left side
   ↪ of middle shelf", or "left side of bottom shelf"
   - "right side of fridge" contains: "right side of top shelf", "right
   ↪ side of middle shelf", or "right side of bottom shelf"
   - "top shelf" contains: "left side of top shelf", "right side of top
   ↪ shelf"
   - "middle shelf" contains: "left side of middle shelf", "right side of
   ↪ middle shelf"
   - "bottom shelf" contains: "left side of bottom shelf", "right side of
   ↪ bottom shelf"

 ## Details about the preferences that you need to output
 A preference is a short paragraph that specifies requirements for each
 ↪ category of grocery items. There must be at least one requirement for
```

↪ each category. The type of requirement for each category can be
↪ different. The categories are: "Fruits", "Vegetables", "Juice-and-soft
↪ -drinks", "Dairy-Products", and "Condiments".

The requirement needs to be one of the following:
- **Type-1. Specific Locations.** These represent that the object must
↪ place at this specific location. The options are:
  - "left side of top shelf"
  - "right side of top shelf"
  - "left side of middle shelf"
  - "right side of middle shelf"
  - "left side of bottom shelf"
  - "right side of bottom shelf".
- **Type-2. General Locations.** These are vaguer locations that contain
↪ multiple specific locations. The options are:
  - "left side of fridge": which means that the user is ok if the object
  ↪ is placed on "left side of top shelf", "left side of middle shelf",
  ↪ or "left side of bottom shelf"
  - "right side of fridge": which means that the user is ok if the object
  ↪ is placed on "right side of top shelf", "right side of middle shelf",
  ↪  or "right side of bottom shelf"
  - "top shelf": which means that the user is ok if the object is either
  ↪ placed on the "left side of top shelf" or "right side of top shelf"
  - "middle shelf": which means that the user is ok if the object is
  ↪ either placed on the "left side of middle shelf" or "right side of
  ↪ middle shelf"
  - "bottom shelf": which means that the user is ok if the object is
  ↪ either placed on the "left side of bottom shelf" or "right side of
  ↪ bottom shelf"
- **Type-3. Relative Positions.** The options are:
  - "<category> must be placed together next to existing <category>
  ↪ regardless of which shelf they are on.": This means that the user
  ↪ only cares that a category of objects are placed together next to
  ↪ existing objects of the same type. The user does not care which
  ↪ specific shelf or which side of the shelf the objects are placed on.
  ↪ For example: "fruits must be placed together next to existing fruits
  ↪ regardless which shelf they are on."
  - "<category> must be placed on the same shelf next to <another category
  ↪  of objects>, and which specific shelf does not matter.": This means
  ↪ that the user only cares that a category of objects are placed
  ↪ together on the same shelf next to another category of objects. It
  ↪ does not matter which specific shelf the objects are on. For example:
  ↪  "fruits must be placed on the same shelf next to condiments, and
  ↪ which specific shelf does not matter."

In addition to giving specific requirements for each category of grocery
↪ items, sometimes you may choose to add additional requirements. The
↪ options are:
- **Type-4. Exception For Attribute**
  - "<category> needs to be placed at <specific location 1>, but <
  ↪ attribute of category> needs to be placed at <specific location 2>.":
  ↪  This means that the user has a different specific requirement for
  ↪ object with a certain attribute compared to the main category. An
  ↪ attribute includes a subcategory of the object, the size/weight of
  ↪ the object, a specific feature of the object, etc. For example, "
  ↪ Dairy product needs to be placed at the right side of top shelf, but
  ↪ cheese needs to be placed at left side of middle shelf." Here, "dairy
  ↪  product" is the main category, and "cheese" is the attribute, which
  ↪ is a specific type of dairy product. Another example is, "Fruits

```
      ↪ needs to be placed at the left side of middle shelf, but big fruits
      ↪ needs to be placed at right side of bottom shelf." Here, "fruits" is
      ↪ the main category, and "big fruit" is the attribute, which is about
      ↪ the size of the fruit.
  - **Type-5. Conditional On Space**
    - "If there are less than <N> objects at <primary specific location>,
    ↪ I want <category> to be placed at <primary specific location>. Else
    ↪ , I want <category> to be placed at <second choice specific location
    ↪ >.": This means that there is a maximum number of objects that can be
    ↪  placed at top choice specific location. If that top choice specific
    ↪ location does not have less than N number of objects, the user wants
    ↪ the a category of objects to be placed at the second choice specific
    ↪ location. For example, "if there are less than 3 objects at the right
    ↪  side of top shelf, I want dairy products to be placed at right side
    ↪ of top shelf. Else, I want dairy products to be placed at left side
    ↪ of middle shelf."

Below is the reasoning worksheet and output that you must follow. <>
↪ contains parts that you must fill out and instructions on how to fill
↪ out that part.

# Reasoning worksheet
## (Fruits) Reasoning about fruits
### (Fruits.A) Summarize where fruits are in the demonstrations
- In demonstration 1:
  - Fruits initially in the fridge:
    - <You must write down an unordered list of fruits that are initially
    ↪ in the fridge. For each fruit, you must write down the specific
    ↪ locations it occupy, and what other grocery items are next to it>
  - Fruits that got placed in the fridge:
    - <You must write down an unordered list of fruits that are initially
    ↪ in the fridge, and the specific locations they occupy>
<You must copy the same format and follow the same process for
↪ demonstration 2.>

### (Fruits.B) When fruits are in the fridge, are they only appearing in
↪ one specific location? If that is the case, what is that specific
↪ location?
<You must look at the summary you wrote in (Fruits.A). You must carefully
↪  analyze each fruit's location in verbose detail. If you answer yes,
↪ you must write down the specific location that fruits are placed at.>

### (Fruits.C) Only if you answered no in (Fruits.B), you should answer
↪ this section. When fruits appear at different locations in the fridge,
↪  you must group the fruits based on their specific locations they
↪ appear at.
<Based on the summary you wrote in (Fruits.A), you must write down a json
↪  (a dictionary) that map each specific location to fruits at that
↪ specific location>
```json
{
  "demonstration 1": {
    "<specific_location_1>": [<You must write down the fruits at
    ↪ specific_location_1 as a list of strings>],
    "<specific_location_2>": [<You must write down the fruits at
    ↪ specific_location_2 as a list of strings>],
    ... <You must do this for each specific location that appeared in your
    ↪  summary in (Fruits.A)>
```

```
    },
    <You must do this for each demonstration.>
  }
‘‘‘
```

### (Fruits.D) Potential preferences for fruits
#### (Fruits.D.1) **Type-1. Specific Locations.**
<Based on section (Fruits.B), what are some specific locations that the
↪ user might like the fruits to be placed at?>

#### (Fruits.D.2) **Type-2. Group Locations.**
<Based on the description of group locations explained at **Type-2.
↪ General Locations.** and the specific locations in mentioned in (
↪ Fruits.A) and (Fruits.C), what are some general locations that the
↪ user might like the fruits to be placed at? For example, if the
↪ specific locations is "left side of top shelf", there are two possible
↪  general locations for fruits: "top shelf" or "left side of the fridge
↪ ". You must write down your reasoning in verbose detail, and you must
↪ specify general locations that fruits might be in.>

#### (Fruits.D.3) **Type-3. Relative Positions.**
<Based on the section (Fruits.A), what type of objects are fruits
↪ typically placed next to? What kind of objects do you think the user
↪ might like the fruits to be placed next to? You must write down your
↪ reasoning in verbose detail.>

#### (Fruits.D.4) **Type-4. Exception For Attribute.**
<Only if your answer in (Fruits.B) is no, you should proceed to answer
↪ this section. Based on the json in (Fruits.C), for each specific
↪ location, what kind of attribute can you identify the fruits at that
↪ specific location share? You must write down your reasoning in verbose
↪  detail.>

#### (Fruits.D.5) **Type-5. Conditional On Space.**
<Only if your answer in (Fruits.B) is no, you should proceed to answer
↪ this section. Based on the json in (Fruits.C), for each specific
↪ location, what kind of conclusion can you make about the number of
↪ objects there? What is a primary location that the user likes to put
↪ fruits at? What is a secondary location that the user would put fruits
↪  at? What is the condition for user to go from putting fruits at the
↪ primary location to the secondary location?>

<You must now copy from ## (Fruits) Reasoning about fruits to #### (
↪ Fruits.D.5) **Type-5. Conditional On Space.** and repeat the reasoning
↪  process for the rest of the categories: ## (Vegetables) Reasoning
↪ about vegetables, ## (Juice-and-soft-drinks) Reasoning about juice and
↪  soft drinks, ## (Dairy-Products) Reasoning about dairy products, ## (
↪ Condiments) Reasoning about condiments. You must write down the
↪ potential preferences for all 5 categories. You cannot omit any
↪ category and you must write down detailed reasoning for all of them. >

--------- System Message --------
You are an assistant who sees an analysis of the demonstrations that the
↪ user have provided and propose potential preferences that the user might
↪  have.
--------- Instruction --------
 # Input
 You are given a reasoning worksheet written in markdown formatthat has
 ↪ already carefully analyzed the demonstrations that the user has

↪ provided to represent their personal preference. You must pay close
↪ attention to this reasoning worksheet.

# Goal
Your goal is to generate <num_preferences> fundamentally different and
↪ diverse preferences that are consistent with the reasoning worksheet
↪ and explain what the user want.

# Instructions and useful information
## Specific locations in the fridge
The fridge can be segmented into 3 shelves (top shelf, middle shelf,
↪ bottom shelf); each shelf has 2 sides (left side, right side). The
↪ fridge has 6 specific locations:
- left side of the top shelf, right side of the top shelf
- left side of the middle shelf, right side of the middle shelf
- left side of the bottom shelf, right side of the bottom shelf

## Details about the preferences that you need to output
A preference is a short paragraph that specifies requirements for each
↪ category of grocery items. There must be at least one requirement for
↪ each category present in the demonstrations. The type of requirement
↪ for each category can be different. The categories are: "Fruits", "
↪ Vegetables", "Juice-and-soft-drinks", "Dairy-Products", and "
↪ Condiments".
If no objects from a given category are present in any of the
↪ demonstrations, you should omit that category from your planning. Do
↪ not randomly choose a rule. Your preferences should also not re-
↪ explain what the categories are or include.

The requirement needs to be one of the following:
- **Type-1. Specific Locations.** These represent that the object must
↪ place at this specific location. The options are:
  - "left side of top shelf"
  - "right side of top shelf"
  - "left side of middle shelf"
  - "right side of middle shelf"
  - "left side of bottom shelf"
  - "right side of bottom shelf".
- **Type-2. General Locations.** These are vaguer locations that contain
↪ multiple specific locations. The options are:
  - "left side of fridge": which means that the user is ok if the object
  ↪ is placed on "left side of top shelf", "left side of middle shelf",
  ↪ or "left side of bottom shelf"
  - "right side of fridge": which means that the user is ok if the object
  ↪ is placed on "right side of top shelf", "right side of middle shelf",
  ↪  or "right side of bottom shelf"
  - "top shelf": which means that the user is ok if the object is either
  ↪ placed on the "left side of top shelf" or "right side of top shelf"
  - "middle shelf": which means that the user is ok if the object is
  ↪ either placed on the "left side of middle shelf" or "right side of
  ↪ middle shelf"
  - "bottom shelf": which means that the user is ok if the object is
  ↪ either placed on the "left side of bottom shelf" or "right side of
  ↪ bottom shelf"
- **Type-3. Relative Positions.** The options are:
  - "<category> must be placed together next to existing <category>
  ↪ regardless of which shelf they are on.": This means that the user
  ↪ only cares that a category of objects are placed together next to
  ↪ existing objects of the same type. The user does not care which

↪ specific shelf or which side of the shelf the objects are placed on.
↪ For example: "fruits must be placed together next to existing fruits
↪ regardless which shelf they are on."
    - "<category> must be placed on the same shelf next to <another category
    ↪  of objects>, and which specific shelf does not matter.": This means
    ↪ that the user only cares that a category of objects are placed
    ↪ together on the same shelf next to another category of objects. It
    ↪ does not matter which specific shelf the objects are on. For example:
    ↪  "fruits must be placed on the same shelf next to condiments, and
    ↪ which specific shelf does not matter."

In addition to giving specific requirements for each category of grocery
↪ items, sometimes you may choose to add additional requirements. The
↪ options are:
- **Type-4. Exception For Attribute**
    - "<category> needs to be placed at <specific location 1>, but <
    ↪ attribute of category> needs to be placed at <specific location 2>.":
    ↪  This means that the user has a different specific requirement for
    ↪ object with a certain attribute compared to the main category. An
    ↪ attribute includes a subcategory of the object, the size/weight of
    ↪ the object, a specific feature of the object, etc. For example, "
    ↪ Dairy product needs to be placed at the right side of top shelf, but
    ↪ cheese needs to be placed at left side of middle shelf." Here, "dairy
    ↪  product" is the main category, and "cheese" is the attribute, which
    ↪ is a specific type of dairy product. Another example is, "Fruits
    ↪ needs to be placed at the left side of middle shelf, but big fruits
    ↪ needs to be placed at right side of bottom shelf." Here, "fruits" is
    ↪ the main category, and "big fruit" is the attribute, which is about
    ↪ the size of the fruit.
- **Type-5. Conditional On Space**
    - "If there are less than <N> objects at <primary specific location>,
    ↪ I want <category> to be placed at <primary specific location>. Else
    ↪ , I want <category> to be placed at <second choice specific location
    ↪ >.": This means that there is a maximum number of objects that can be
    ↪  placed at top choice specific location. If that top choice specific
    ↪ location does not have less than N number of objects, the user wants
    ↪ the a category of objects to be placed at the second choice specific
    ↪ location. For example, "if there are less than 3 objects at the right
    ↪  side of top shelf, I want dairy products to be placed at right side
    ↪ of top shelf. Else, I want dairy products to be placed at left side
    ↪ of middle shelf."

Below is the output that you must follow. <> contains parts that you must
↪  fill out and instructions on how to fill out that part. You must only
↪  copy and output what is under # Potential Preference Generation
↪ Process.

# Potential Preference Generation Process.
## (I) Propose potential preferences based reasoning worksheet
<You must write <num_preferences> potential preferences that are diverse
↪ in the requirements for each category. The potential preferences
↪ CANNOT only differ in the wording they have. They must be
↪ fundamentally different with different requirements for at least one
↪ object. >
### (I.1) Potential preference 1
#### (I.1.a) What type of requirements are you picking for each category?
↪  You must pick different types of requirements compared to previous
↪ preferences, so how are types that you picked different from previous
↪ ones?

```
<For each category, you must pick at least one or more type of
↪ requirements based on your answers in (Fruits.D), (Vegetables.D), (
↪ Juice-and-soft-drinks.D), (Dairy-Products.D), (Condiments.D). The type
↪  of requirements for each category can be the same, but they do not
↪ have to be the same for all categories. For each category, you must
↪ verbosely explain in detail what is the type of requirement that you
↪ picked and what exactly is the preference for that category. If this
↪ is the first potential preference that you are writing, you do not
↪ need to answer how the types that you picked are different from
↪ previous preferences. However, for subsequent potential preferences
↪ that you will write, you must make sure that you are picking a
↪ different type of requirements for the 5 categories. You must explain
↪ in detail how the types that you pick are different from previous
↪ preferences. You must cite the previous potential preferences by their
↪  number, for example (I.1.b).>

#### (I.1.b) Preference
<Based on the types of requirements that you picked in (I.1.a), your goal
↪  is to write in a paragraph a potential user preference that is
↪ consistent with the demonstrations and the reasoning worksheet. You
↪ must write in natural langauge and there must be at least one
↪ requirement for each category. You must not make reference to other
↪ potetial preferences or reasoning that you have written. Each
↪ preference be understandable on their own without referring to another
↪  potential preference. >

<You must now copy from ### (I.1) Potential preference 1 and repeat the
↪ writing process for the remaining <num_preferences_minus_one>
↪ potential preferences. You must write <num_preferences> potential
↪ preferences in total.>

# (II) Final valid JSON output
You must copy the <num_preferences> diverse potential preferences that
↪ you wrote in (II) as a list of strings in a valid json format. You
↪ must only copy the preference, which must not refer to other
↪ preferences or parts of the writing process. Each preference must
↪ define requirements for all the categories, and it must be
↪ understanable and interpretable on its own without referring to other
↪ potential preferences or other writing that you have done before. Each
↪  preference should You must make sure that when writing each
↪ preference, you do not use single quotes or double quotes or citations
↪ .
```json
[
    "<You must copy the first potential preference in (I.1.b) here. For
    ↪ subsequent potential preferences, you must not make reference to
    ↪ previous potential preferences.>",
    <You must copy the remaining <num_preferences_minus_one> preferences
    ↪ here>
]
```
```

### 9.1.2 Calculate the Reward

The LLM is prompted to output a json of yes/no on whether an object's semantic placement location in a plan $\xi$ satisfies a given preference $\theta$. The reward is calculated by the ratio between the sum of the responses (yes is 1 and no is 0) and the total number of objects in the plan. We use gpt-4o for this prompt.

```
--------- System Message --------
You are roleplaying as a user with a specific preference. Your goal is to
↪ answer whether or not an object placement has satisfied your preference
↪ and the initial configuration of the fridge.
--------- Instruction --------
  You are roleplaying as a user with a specific preference. You are shown
  ↪ the objects that are already initially in the fridge. The fridge has
  ↪ different shelves, which are divided into smaller sub-sections. The
  ↪ objects in a specific section are listed from left to right.

  You are also shown a plan to place an object at a specific part of the
  ↪ fridge. Your goal is to answer "yes" or "no" on whether the object
  ↪ placement has satisfied your preference. You must also pay attention
  ↪ to objects that are already initially in the fridge because where you
  ↪ want objects placed could depend that initial condition.

  You must respect the following rules when you make your decision:
  - You do not need to worry about the amount of space in the fridge. You
  ↪ must not make your decision based on you judging if a shelf is already
  ↪  full.
  - If the preference for a category is a specific location, you must only
  ↪ check if the object is being placed at that specific location. You
  ↪ must ignore other objects that are at that specific location.
    * For example, consider when your preference is to put drinks together
    ↪ regardless of the shelf and to put fruits on the left side of top
    ↪ shelf, and the plan has placed a drink on the left side of the top
    ↪ shelf then a fruit also on the left side of top shelf. When you
    ↪ answer whether the plan of placing the fruit on the left side of top
    ↪ shelf, you should answer "yes" because it satisfies the specific
    ↪ placement location for fruit.
  - If the object placement plan is more general than your preference, that
  ↪  plan does not satisfy your preference, and you must reply "no".
    * For example, if your preference is to put fruits on the left side of
    ↪ the middle shelf, and the plan tries to put a fruit on the middle
    ↪ shelf, the plan is too general, and it will not always satisfy your
    ↪ preference. Thus, you must reply "no".
  - When your preference has conditionals, you must carefully check if that
  ↪  conditional is true first when you make your decision.
    * For example, if your preference wants vegetables to be at location 2
    ↪ if there are already N objects at location 1, you must check the
    ↪ objects that are already intially in the fridge and make sure that
    ↪ the condition (N objects at location 1) is true.
  - When your preference depends on the initial condition of the fridge,
  ↪ you must carefully check what is desirable placement location based on
  ↪  the inital condition of the fridge.
    * For example, if your preference wants vegetables to be placed together
    ↪ , you must check if there are already vegetables in the fridge. If
    ↪ there are already vegetables at the right side of top shelf, the
    ↪ object placement plan should also place new objects at the right side
    ↪  of top shelf.

  The preference, the objects already initially in the fridge, the object
  ↪ placement plan that you receive will be in markdown format:
  # Your preference
  ... Your preference will be stated here ...

  # Objects already initially in the fridge
  ‘‘‘
```

```
... The objects already initially in the fridge will be stated here as an
↪ json ...
```

# Object placement plan
```
... The plan will be stated here as a python code: pickandplace(
↪ object_to_place, placement_location)
```

You must reply in a valid json format, each item corresponds to one of
↪ the pickandplace() function in the object placement plan. You must
↪ only use double quote. You cannot use apostrophy or single quote. See
↪ below for the format that you must follow.
```json
[
  {
      "Reasoning": "{You must not use any more of quotation mark inside
      ↪ your resoning. You must not write apostrophe, double quote, or
      ↪ single quote. You must put your reasoning on whether a user with
      ↪ your preference would be happy with the object placement given
      ↪ the objects already initially in the fridge.}",
      "Does this plan satisfy your preference (yes/no)": "{you must only
      ↪ reply yes or no}"
  }, ...<there should be one dictionary per line of pickandplace()>
]
```
--------- Example Input --------
   # Your preference
   I like putting dairy on the top shelf and vegetables on the left side of
   ↪  the bottom shelf.

   # Objects already initially in the fridge
   ```
   {
       "top shelf":
           {
               "left side of top shelf": [],
               "right side of top shelf": ["cheese", "yogurt"]
           },
       "middle shelf":
           {
               "left side of middle shelf": [],
               "right side of middle shelf": []
           },
       "bottom shelf":
           {
               "left side of bottom shelf": ["carrot"],
               "right side of bottom shelf": []
           }
   }
   ```

   # Object placement plan
   ```
   pickandplace("oat milk", "right side of top shelf")
   pickandplace("whole milk", "right side of middle shelf")
   ```
--------- Example Response --------

```json
[
    {
        "Reasoning": "Oat milk is a dairy product. I prefer dairy on the
        ↪  top shelf. The plan was to put oat milk on the right side of
        ↪  top shelf, which falls under top shelf, so the plan
        ↪ satisfies my preference.",
        "Does this plan satisfy your preference (yes/no)": "yes"
    },
    {
        "Reasoning": "Whole milk is a dairy product. I prefer dairy on
        ↪ the top shelf. The plan was to put whole milk on the right
        ↪ side of middle shelf, which is not top shelf, so the plan
        ↪ does not satisfy my preference.",
        "Does this plan satisfy your preference (yes/no)": "no"
    }
]
```
--------- Example Input --------
# Your preference
I like fruits on the left side of the middle shelf, and vegetables on
↪ the right side of the middle shelf.

# Objects already initially in the fridge
```
{
    "top shelf":
        {
            "left side of top shelf": ["whole milk"],
            "right side of top shelf": []
        },
    "middle shelf":
        {
            "left side of middle shelf": [],
            "right side of middle shelf": []
        },
    "bottom shelf":
        {
            "left side of bottom shelf": [],
            "right side of bottom shelf": []
        }
}
```

# Object placement plan
```
pickandplace("apple", "middle shelf")
```
--------- Example Response --------
```json
[
    {
        "Reasoning": "Apple is a fruit. I prefer fruits on the left side
        ↪  of the middle shelf. The plan is placing apple in a more
        ↪ general location: middle shelf. This means that it could
        ↪ potentially place apple not specifically at the left side of
        ↪ the middle shelf, so the plan does not satisfy my preference
        ↪ .",
        "Does this plan satisfy your preference (yes/no)": "no"
```

```
        }
    ]
    ```
--------- Example Input --------
    # Your preference
    I prefer vegetables be placed together. I also want fruits on the middle
    ↪  shelf.

    # Objects already initially in the fridge
    ```
    {
        "top shelf":
            {
                "left side of top shelf": [],
                "right side of top shelf": []
            },
        "middle shelf":
            {
                "left side of middle shelf": [],
                "right side of middle shelf": ["cucumber", "carrot"]
            },
        "bottom shelf":
            {
                "left side of bottom shelf": [],
                "right side of bottom shelf": []
            }
    }
    ```

    # Object placement plan
    ```
    pickandplace("spinach", "right side of top shelf")
    ```
--------- Example Response --------
    ```json
    [
        {
            "Reasoning": "Spinach is a vegetable. I want vegetables to be
            ↪ placed together, so I need to check if there are vegetables
            ↪ initially in the fridge. There are vegetables (cucumber,
            ↪ carrot) at the right side of middle shelf, so new vegetables
            ↪ should also get placed at the right side of middle shelf.
            ↪ However, the plan decides to place spinach to the right side
            ↪ of the top shelf, which is not right side of middle shelf.
            ↪ Thus, this plan does not satisfy my preference.",
            "Does this plan satisfy your preference (yes/no)": "no"
        }
    ]
    ```
--------- Example Input --------
    # Your preference
    If there are less than 2 dairy product on the right side of top shelf,
    ↪ you can put dairy products on the right side of top shelf; else, I
    ↪ want the rest of the dairy product to be on the right side of middle
    ↪ shelf.

    # Objects already initially in the fridge
    ```
    {
```

```
        "top shelf":
            {
                "left side of top shelf": [],
                "right side of top shelf": ["oat milk"]
            },
        "middle shelf":
            {
                "left side of middle shelf": [],
                "right side of middle shelf": []
            },
        "bottom shelf":
            {
                "left side of bottom shelf": [],
                "right side of bottom shelf": []
            }
    }
    ```

    # Object placement plan
    ```
    pickandplace("whole milk", "right side of middle shelf")
    ```
--------- Example Response --------
    ```json
    [
        {
            "Reasoning": "Whole milk is a dairy product. My preference has a
            ↪  condition: whether there are less than 2 dairy products on
            ↪ the right side of top shelf, so I must pay close attention to
            ↪  the objects initially in the fridge. There is 1 dairy
            ↪ product (oat milk), so the condition (less than 2 dairy
            ↪ product on the right side of top shelf) is true. My
            ↪ preference would want whole milk also on the right side of
            ↪ top shelf. However, the plan put whole milk on right side of
            ↪ middle shelf, so it does not satisfy my preference.",
            "Does this plan satisfy your preference (yes/no)": "no"
        }
    ]
    ```
--------- Example Input --------
    # Your preference
    If there are less than 2 dairy product on the right side of top shelf,
    ↪ you can put dairy products on the right side of top shelf; else, I
    ↪ want the rest of the dairy product to be on the right side of middle
    ↪ shelf.

    # Objects already initially in the fridge
    ```
    {
        "top shelf":
            {
                "left side of top shelf": [],
                "right side of top shelf": ["oat milk", "whole milk"]
            },
        "middle shelf":
            {
                "left side of middle shelf": [],
                "right side of middle shelf": []
            },
```

```
            "bottom shelf":
                {
                    "left side of bottom shelf": [],
                    "right side of bottom shelf": []
                }
        }
    ```

    # Object placement plan
    ```
    pickandplace("cheese", "right side of top shelf")
    ```
--------- Example Response --------
    ```json
    [
        {
            "Reasoning": "Cheese is a dairy product. My preference has a
            ↪ condition: whether there are less than 2 dairy products on
            ↪ the right side of top shelf, so I must pay close attention to
            ↪  the objects initially in the fridge. There are 2 dairy
            ↪ product (oat milk, whole milk), so the condition (less than 2
            ↪  dairy product on the right side of top shelf) is false. I
            ↪ should look at the else case of my prefernece, which wants
            ↪ the cheese to be on the right side of middle shelf. However,
            ↪ the plan put cheese on right side of top shelf, so it does
            ↪ not satisfy my preference.",
            "Does this plan satisfy your preference (yes/no)": "no"
        }
    ]
    ```
--------- Example Input --------
    # Your preference
    I prefer vegetables be placed on the right side of middle shelf. I want
    ↪ fruits on the right side of top shelf, condiments on the left side of
    ↪  top shelf.

    # Objects already initially in the fridge
    ```
    {
        "top shelf":
            {
                "left side of top shelf": [],
                "right side of top shelf": []
            },
        "middle shelf":
            {
                "left side of middle shelf": [],
                "right side of middle shelf": ["cucumber", "carrot", "corn"]
            },
        "bottom shelf":
            {
                "left side of bottom shelf": [],
                "right side of bottom shelf": []
            }
    }
    ```

    # Object placement plan
    ```
```

```
    pickandplace("spinach", "right side of bottom shelf")
    ```
--------- Example Response --------
    ```json
    [
        {
            "Reasoning": "Spinach is a vegetable. I prefer vegetables on the
            ↪  right side of middle shelf. The plan is placing spinach at
            ↪ the wrong location: right side of bottom shelf. Thus, this
            ↪ plan does not satisfy my preference.",
            "Does this plan satisfy your preference (yes/no)": "no"
        }
    ]
    ```
--------- Example Input --------
    # Your preference
    I prefer vegetables be placed together next to exsiting vegetables
    ↪ regardless of which shelf they are on. I want fruits on the right
    ↪ side of top shelf.

    # Objects already initially in the fridge
    ```
    {
        "top shelf":
            {
                "left side of top shelf": [],
                "right side of top shelf": ["cucumber"]
            },
        "middle shelf":
            {
                "left side of middle shelf": [],
                "right side of middle shelf": []
            },
        "bottom shelf":
            {
                "left side of bottom shelf": [],
                "right side of bottom shelf": []
            }
    }
    ```

    # Object placement plan
    ```
    pickandplace("spinach", "right side of top shelf")
    pickandplace("apple", "right side of top shelf")
    ```
--------- Example Response --------
    ```json
    [
        {
            "Reasoning": "Spinach is a vegetable. I want vegetables to be
            ↪ placed together, so I need to check if there are vegetables
            ↪ initially in the fridge. There are vegetables (cucumber) at
            ↪ the right side of top shelf, so new vegetables should also
            ↪ get placed at the right side of top shelf. Since the plan put
            ↪  spinach at the top shelf, it has satisfied my preference.",
            "Does this plan satisfy your preference (yes/no)": "yes"
        },
        {
```

```
            "Reasoning": "Apple is a fruit. I prefer fruits on the right
         ↪ side of top shelf. The plan is placing apple at the right
         ↪ side of top shelf, which matches the requirement and
         ↪ satisfies my preference.",
            "Does this plan satisfy your preference (yes/no)": "yes"
        }
    ]
    ‘‘‘
```

### 9.1.3   Generate Candidate Questions

The LLM receives as input a pair of preferences, the corresponding plans for these preferences, the reward of each plan given these preferences, the initial condition, and the tasks that these plans are generated to solve. We use gpt-4o for this prompt.

```
--------- System Message --------
You are an assistant that ask informative yes-or-no questions to try to
↪ differentiate between two potential preference candidates.
--------- Instruction --------
  # Goal
  You are trying to learn what is a user's personal preference. You
  ↪ receivea new initial state that you are trying to solve, two possible
  ↪ candidates of the user's preference, the corresponding task plan
  ↪ generated based on the candidate preferences.

  Your goal is to analyze the differences between the two candidates and
  ↪ their plans and ask <num_questions> informative, effective yes-or-no
  ↪ questions such that the user's answers to your questions can reveal
  ↪ which candidate is closer to the user's preference.

  # Input
  You will receive the input in the following format:
  # New Initial Condition To Solve
  ## Objects initially in the fridge
  ‘‘‘json
  ... The objects already initially in the fridge will be stated here as an
  ↪  json ...
  ‘‘‘

  ## Objects that must be put away into the fridge
  ‘‘‘json
  ... The objects that need to be put away into the fridge will be stated
  ↪ here as a list ...
  ‘‘‘

  # Preference Candidate 1
  ... The first candidate will be stated here ...

  ## Plan 1 based on Preference Candidate 1
  ... The plan will put away all the objects that need to be put away ...

  ### Preference Candidate 1's Score on Plan 1
  ... Numerical score here. The higher a score, the better Plan 1 is at
  ↪ satisfying the preference candidate 1...
  ### Preference Candidate 2's Score on Plan 1
  ... Numerical score here. The higher a score, the better Plan 1 is at
  ↪ satisfying the preference candidate 2 ...

  # Preference Candidate 2
```

```
... The second candidate will be stated here ...

## Plan 2 based on Preference Candidate 2
... The plan will put away all the objects that need to be put away ...

### Preference Candidate 1's Score on Plan 2
... Numerical score here. The higher a score, the better Plan 2 is at
↪ satisfying the preference candidate 1...
### Preference Candidate 2's Score on Plan 2
... Numerical score here. The higher a score, the better Plan 2 is at
↪ satisfying the preference candidate 2 ...

You must follow these rules when coming up with questions to ask:
- You must analyze the difference in the candidate preferences and their
↪ plans. The differences will inform you on what question to ask. The
↪ scores can inform you how much each plan is liked by each preference
↪ candidate. The scores can help guide you to understand the main
↪ difference in the candidate preferences.
- You must ask questions about the preferences. You must never ask
↪ question about that mention the new initial condition to solve or the
↪ plan that got generated.
- Your questions must just be about what the user prefer. They must be
↪ standalone so that the user can just look at the question and be able
↪ to answer that question.
- You must ask effective, useful questions where user's answer will help
↪ you figure out which preference candidate is better.
- You must ask simple yes or no questions.
- Your questions cannot ask if the user prefers A or B. However, you can
↪ ask if the user prefers A rather than B.
- Your questions should try to use wording or terminology that are also
↪ used in the candidate preferences.
- Your question must be about specifc categories or attributes under a
↪ specific category. The categories are for example: ["fruits", "
↪ vegetables", "dairy product", "condiments", "juice and soft drinks"].
↪ Some examples of attributes are: "heavy fruits", which is an attribute
↪  under "fruits"; "sweet conditments", which is an attribute under "
↪ condiments"; "soft drinks", which is an attribute under "juice and
↪ soft drinks".
- Your questions must not be about overall objects or items. You must
↪ never ask questions like "Do you prefer to items to be ...?" or "Do
↪ you prefer items of the same category ...?"
- Your questions must not be asking about specific objects. You must
↪ never ask questions like "Do you prefer chocolate milk to be placed at
↪  ...?" Instead, you can ask questions about the specific category (e.g
↪ . "Do you prefer dairy products to be placed at ...?" or specific
↪ attribute (e.g. "Do you prefer milk to be placed at", where "milk" is
↪ the specific attribute.)

You must reply in this format:
# Reasoning
 You should analyze the differences between preference candidates and
↪ their plans. You should strategize what questions you would ask here

# Questions
 You must put your <num_questions> questions as an unordered list here
```

### 9.1.4 Answering a Question Given a Preference

The LLM is asked to roleplay a user who has a preference $\theta$ answering the question $q$. We use `gpt-4o` for this prompt. This prompt is used to:

- Estimate the likelihood $P(o|q, \theta)$: We use OpenAI's API, which outputs the log-likelihood of outputting a specific token, to extract the log-likelihood of the LLM outputting "yes" or "no".

- Mimic the user's answer with ground-truth preference: When the interactive approaches need to query the user, the LLM is supplied with the ground-truth preference (not known to the approaches) to answer the question that an approach asks.

```
--------- System Message --------
You are roleplaying as a user with a specific preference. Your goal is to
↪ answer yes-or-no question based on the preference of the user whom you
↪ are roleplaying. When you put your answer in the json, you must only
↪ write "yes" or "no"
--------- Instruction --------
 You are roleplaying as a user with a specific preference. You are asked
 ↪ one yes-or-no question, and you goal is answer "yes" or "no" based on
 ↪ your preference.

 The preference and the questions that you receive will be in markdown
 ↪ format:
 # Your preference
  Your preference will be stated here

 # Question for you to answer
  The question will be stated here as an unordered list

 You must follow these rules when answering:
 - When you write down your reasoning, you must not use apostrophy, single
 ↪  quote, or double quote.
 - You must only answer "yes" or "no". Even if it does not apply to your
 ↪ preference or you are not sure, you must answer "yes" or "no". You
 ↪ cannot answer anything else. You must not write "N/A" or "na" or "n/a
 ↪ ". You must only output "yes" or "no".
 - If the question is asking whether the user prefers a specific thing,
 ↪ but the user's preference is more general, the user will likely
 ↪ respond no because they do not care about the specific thing and they
 ↪ only care about the general requirement.
 - If the question is asking whether the user prefers a general thing, but
 ↪  the user's preference is more specific, the user will likely respond
 ↪ no because the general thing can include cases that violate the user's
 ↪  specific requirement.

 You must reply in a json, which includes your reasoning process and
 ↪ answer to a question. You must follow this format:
 {
   "Reasoning": "{put your reasoning on how a user with your preference
   ↪ would answer to this question. You must not use single quote or
   ↪ double quote inside your reasoning. }",
   "Answer (yes/no)": "{you must only reply yes or no}"
 }
--------- Example Input --------
   # Your preference
   Fruits should be placed on left side of the fridge.

   # Questions for you to answer
```

```
        - Do you prefer fruits to be placed on the left side of the top shelf?
--------- Example Response --------
    {
      "Reasoning": "The user prefers fruits to be on the left side of the
      ↪ fridge, which is a general requirement. The user does not
      ↪ specifically perfer fruits to be on the left side of the top shelf,
      ↪  so the user would answer no.",
      "Answer (yes/no)": "no"
    }
--------- Example Input --------
    # Your preference
    Fruits should be placed on left side of the fridge.

    # Questions for you to answer
    - Is it important to you to have fruits specifically on the left side of
    ↪  the middle shelf rather than just on the right left of the fridge?
--------- Example Response --------
    {
      "Reasoning": "The user prefers fruits to be on the left side of the
      ↪ fridge, which is a general requirement and the user will not care
      ↪ about the specific shelf that fruits are placed on as long as it is
      ↪  the left side of the fridge. Thus, it is not important for fruits
      ↪ to be on the left side of the middle shelf, and the user would
      ↪ answer no.",
      "Answer (yes/no)": "no"
    }
--------- Example Input --------
    # Your preference
    Fruits should be placed on left side of the top shelf.

    # Questions for you to answer
    - Do you prefer fruits to be placed on the left side of the fridge?
--------- Example Response --------
    {
      "Reasoning": "The user prefers that fruits are on the left side of the
      ↪  top shelf, which is specific. Left side of the fridge includes
      ↪ left side of the middle shelf and left side of the bottom shelf,
      ↪ which does not match preference of the user. The user would answer
      ↪ no.",
      "Answer (yes/no)": "no"
    }
--------- Example Input --------
    # Your preference
    Fruits should be placed on the left side of the middle or bottom shelf.

    # Questions for you to answer
    - Do you prefer to place fruits next to other fruits already in the
    ↪ fridge?
--------- Example Response --------
    {
      "Reasoning": "The user prefers that fruits are placed at specific
      ↪ locations (left side of the middle or bottom shelf). The question
      ↪ is more explicitly asking if the user prefers fruits to be placed
      ↪ next to other fruits already in the fridge, which does not have
      ↪ specfic locations requirement, so the user would say no.",
      "Answer (yes/no)": "no"
    }
--------- Example Input --------
    # Your preference
```

```
    Fruits should be placed on the left side of the middle or bottom shelf.

    # Questions for you to answer
    - Do you prefer to have similar items grouped together on the same shelf
    ↪ ?
--------- Example Response --------
    {
      "Reasoning": "The user prefers that fruits are placed at specific
      ↪ locations (left side of the middle or bottom shelf). The question
      ↪ is implying that the user only cares about similar items being on
      ↪ the same shelf without specific requirements on which shelf and
      ↪ which side of the shelf, so the user would answer no.",
      "Answer (yes/no)": "no"
    }
--------- Example Input --------
    # Your preference
    I want fruits to be placed together next to fruits that are already in
    ↪ the fridge.

    # Questions for you to answer
    - Do you prefer to place fruits next to other fruits already in the
    ↪ fridge?
--------- Example Response --------
    {
      "Reasoning": "The user prefers that fruits are placed together next to
      ↪  fruits that are already in the fridge. The question also asks if
      ↪ the user prefers to place fruits next to other fruits already in
      ↪ the fridge, so the answer is yes.",
      "Answer (yes/no)": "yes"
    }
--------- Example Input --------
    # Your preference
    Fruits should be placed on the left side of the middle shelf.

    # Questions for you to answer
    - Do you prefer if fruits are placed specifically on the middle shelf?
--------- Example Response --------
    {
      "Reasoning": "The user prefers that fruits are placed at specific
      ↪ locations (left side of the middle shelf). The question is asking
      ↪ about a more general requirement because middle shelf is more
      ↪ general than the left side of middle shelf. Based on the rules
      ↪ above, the general location contains locations that violate the
      ↪ user's preference (right side of the middle shelf), so answer is no
      ↪ .",
      "Answer (yes/no)": "no"
    }
--------- Example Input --------
    # Your preference
    Fruits should be placed on the left side of the middle shelf. Vegetables
    ↪  should be on the right side of middle shelf.

    # Questions for you to answer
    - Do you prefer to have items of the same category placed together on
    ↪ the same shelf?
--------- Example Response --------
    {
      "Reasoning": "The user prefers that fruits are placed at specific
      ↪ locations (left side of the middle shelf) and vegetables are also
```

```
        ↪ placed at specific locations (right side of the middle shelf). The
        ↪ question is asking if there is any category of objects that needs
        ↪ to be placed together on the same shelf. Since the requirements for
        ↪  fruits is about a specific location instead of putting fruits
        ↪ together, and the requirements for vegetables is also about a
        ↪ specific location instead of putting vegetables togetheer, the
        ↪ answer to this question is no",
      "Answer (yes/no)": "no"
    }
--------- Example Input --------
   # Your preference
   Fruits should be placed on the left side of the middle shelf. Condiments
   ↪  on the right side of the top shelf. Vegetables should be placed
   ↪ together next to other existing vegetables.

   # Questions for you to answer
   - Do you prefer to have items of the same category placed together on
   ↪ the same shelf?
--------- Example Response --------
   {
      "Reasoning": "The user prefers that fruits are placed at specific
      ↪ locations (left side of the middle shelf), condiments are placed at
      ↪  specific locations (right side of the top shelf), and vegetables
      ↪ need to be placed together. The question asks if there is any
      ↪ category of objects that needs to be placed together on the same
      ↪ shelf. In this preference, vegetables need to be placed together on
      ↪  the same shelf. Since there exists one category that are placed
      ↪ together on the same shelf, the answer to this question is yes",
      "Answer (yes/no)": "yes"
    }
```

## 9.2 Task Planner Prompts

The LLM prompt will generate a semantic plan that satisfies the given user preferences. When it receives feedback, it will reflect on its plan and revise its original plan. We use `gpt-4-turbo` because we empirically find that `gpt-4o` tends to make more mistakes (typos) when generating the code.

```
--------- System Message --------
You are an assistant that comes up with a plan for putting items into a
↪ fridge given a list of items and a human's preference.
--------- Instruction --------
 You must analyze a human's preferences and then come up with a plan to
 ↪ put items into a fridge.

 You will receive the following as input:
 Optional[Feedback]: ...
 Objects: ...
 Locations: ...
 Initial State: ...
 Preference: ...

 where
   - "Feedback" appears if the previous plan was geometrically infeasible.
     - It will contain the items that did not fit in the previous plan.
     - It is possible that the item could fit but the pickandplace function
     ↪  call was incorrect (e.g. misspelled item or location)
   - "Objects" is a list of items that need to be placed in the fridge.
```

```
    - "Locations" is a list of locations in the fridge.
    - "Initial State" is a dictionary whose keys are the locations in the
    ↪ fridge and the values are a list of items currently in that location.
    - "Preference" is a description of the human's preferences for where
    ↪ they like things in a fridge.
      - The preference should always be satisfied, even when attempting to
      ↪ place items that did not fit in the previous plan.

 You must respond in the following format:
 # Reflect: ...
 # Reasoning: ...
 pickandplace(item1, location1)
 pickandplace(item2, location2)
 ...

 where
   - "Reflect" should contain reasoning about the previous plan if it was
   ↪ geometrically infeasible.
     - Reflect on what went wrong and how you plan to fix it.
     - The plan must abide by the human's preference.
   - "Reasoning" should contain the reasoning for your plan.
     - Reason about how best to place the items in the fridge based on the
     ↪ human's preference.
     - If the reflection involves repositioning items, ensure that the
     ↪ human's preference is still satisfied.
   - "pickandplace(item, location)" is a function call that places the item
   ↪  in the location in the fridge.
     - This is your plan of action.

 Each time you are prompted to generate a new plan, the fridge is reset to
 ↪  its initial state. Use this
 to your advantage to come up with a better plan. It is absolutely
 ↪ necessary to satisfy the human's
 preference; geometric infeasibility only suggests that objects should be
 ↪ placed in different locations
 that still satisfy the preference.
--------- Example Input --------
   Objects: ["milk", "cheese", "apple", "orange"]
   Locations: ["top shelf", "left side of top shelf", "right side of top
   ↪ shelf", "middle shelf", "left side of middle shelf", "right side of
   ↪ middle shelf", "bottom shelf", "left side of bottom shelf", "right
   ↪ side of bottom shelf"]
   Initial State: {}
   Preference: "I like putting dairy on the top shelf and fruits on the
   ↪ right side of middle shelf."
--------- Example Response --------
   # Reasoning: Milk and cheese are dairy product, which based on the
   ↪ preference needs to be on the top shelf. Apples and oranges are
   ↪ fruits, which needs to be on the right side of middle shelf.
   pickandplace("milk", "top shelf")
   pickandplace("cheese", "top shelf")
   pickandplace("apple", "right side of middle shelf")
   pickandplace("orange","right side of middle shelf")
--------- Example Input --------
   Objects: ["milk", "cheese", "apple", "orange"]
   Locations: ["top shelf", "left side of top shelf", "right side of top
   ↪ shelf", "middle shelf", "left side of middle shelf", "right side of
   ↪ middle shelf", "bottom shelf", "left side of bottom shelf", "right
   ↪ side of bottom shelf"]
```

```
Initial State: {"left side of middle shelf": ["peach", "cherries"]}
Preference: "Fruits must be placed next to other exisiting fruits
↪ regardless of which shelf they are on. Dairy products on the be the
↪ right side of bottom shelf. "
--------- Example Response --------
# Reasoning: Apples and oranges are fruits, which needs to be placed
↪ next to other exisiting fruits. Existing fruits in the fridge are "
↪ peach" and "cherries", which are at the left side of middle shelf, so
↪  apple and oranges should be placed there as well. Milk and cheese
↪ are diary product, which needs to be placed on the right side of
↪ bottom shelf.
pickandplace("apple", "left side of middle shelf")
pickandplace("orange","left side of middle shelf")
pickandplace("milk", "right side of bottom shelf")
pickandplace("cheese", "right side of bottom shelf")
--------- Example Input --------
Objects: ["milk", "cheese", "apple", "orange"]
Locations: ["top shelf", "left side of top shelf", "right side of top
↪ shelf", "middle shelf", "left side of middle shelf", "right side of
↪ middle shelf", "bottom shelf", "left side of bottom shelf", "right
↪ side of bottom shelf"]
Initial State: {"right side of top shelf": ["cabbage", "corn"]}
Preference: "Fruits must be placed next to other exisiting vegetables
↪ regardless of which shelf they are on. Dairy products on the be the
↪ right side of bottom shelf. "
--------- Example Response --------
# Reasoning: Apples and oranges are fruits, which needs to be placed
↪ next to other exisiting vegetables. Existing vegetables in the fridge
↪  are "cabbage" and "corn", which are at the right side of top shelf,
↪ so apple and oranges should be placed there as well. Milk and cheese
↪ are diary product, which needs to be placed on the right side of
↪ bottom shelf.
pickandplace("apple", "right side of top shelf")
pickandplace("orange","right side of top shelf")
pickandplace("milk", "right side of bottom shelf")
pickandplace("cheese", "right side of bottom shelf")
--------- Example Input --------
Objects: ["milk", "cheese", "apple", "orange"]
Locations: ["top shelf", "left side of top shelf", "right side of top
↪ shelf", "middle shelf", "left side of middle shelf", "right side of
↪ middle shelf", "bottom shelf", "left side of bottom shelf", "right
↪ side of bottom shelf"]
Initial State: {"left side of middle shelf": ["peach", "cherries"]}
Preference: "Most dairy product should be placed on the left side of top
↪  shelf, but cheese product should be placed on the right side of
↪ middle shelf. Fruits should be placed on the left side of middle
↪ shelf."
--------- Example Response --------
# Reasoning: Cheese is a cheese product, so it needs to be placed at
↪ right side of middle shelf. Milk is a dairy product, so it should be
↪ placed on the left side of top shelf. Apples and oranges are fruits,
↪ which needs to be on the left side of middle shelf.
pickandplace("cheese", "right side of middle shelf")
pickandplace("milk", "left side of top shelf")
pickandplace("apple", "left side of middle shelf")
pickandplace("orange","left side of middle shelf")
--------- Example Input --------
Objects: ["milk", "cheese", "apple", "orange"]
```

```
    Locations: ["top shelf", "left side of top shelf", "right side of top
    ↪ shelf", "middle shelf", "left side of middle shelf", "right side of
    ↪ middle shelf", "bottom shelf", "left side of bottom shelf", "right
    ↪ side of bottom shelf"]
    Initial State: {"right side of bottom shelf": ["yogurt", "butter"]}
    Preference: "If the right side of bottom shelf has less than 3 items,
    ↪ dairy products can be placed there. Else, you must place them at the
    ↪ left side of top shelf. Fruits should be placed on the left side of
    ↪ middle shelf."
--------- Example Response --------
    # Reasoning: Right side of bottom shelf can fit one more dairy product,
    ↪ so I will put milk there. Since now there are 3 items at the right
    ↪ side of bottom shelf, I must put cheese on the left side of top shelf
    ↪ . Apples and oranges are fruits, which needs to be on the left side
    ↪ of middle shelf.
    pickandplace("milk", "right side of bottom shelf")
    pickandplace("cheese", "left side of top shelf")
    pickandplace("apple", "left side of middle shelf")
    pickandplace("orange","left side of middle shelf")
--------- Example Input --------
    Objects: ["milk", "cheese", "apple", "melon"]
    Locations: ["top shelf", "middle shelf", "bottom shelf"]
    Initial State: {
      "top shelf": ["yogurt", "butter"],
      "middle shelf": ["watermelon", "pizza box"]
    }
    Preference: "I don't want any of the fridge shelves to be too crowded."
--------- Example Response --------
    pickandplace("milk", "top shelf")
    pickandplace("cheese", "top shelf")
    pickandplace("apple", "middle shelf")
    pickandplace("melon", "middle shelf")
--------- Example Input --------
    Feedback: The previous plan was geometrically infeasible. The items that
    ↪ did not fit were ["melon", "apple"].
    Objects: ["milk", "cheese", "apple", "melon"]
    Locations: ["top shelf", "middle shelf", "bottom shelf"]
    Initial State: {
      "top shelf": ["yogurt", "butter"],
      "middle shelf": ["watermelon", "pizza box"]
    }
    Preference: "I don't want any of the fridge shelves to be too crowded."
--------- Example Response --------
    # Reflect: The melon and apple did not fit in the previous plan. This
    ↪ must mean that the middle shelf is too crowded.
    # Reasoning: Instead of placing the apple and melon on the middle shelf,
    ↪ they can be placed on the bottom shelf which is empty.
    pickandplace("milk", "top shelf")
    pickandplace("cheese", "top shelf")
    pickandplace("apple", "bottom shelf")
    pickandplace("melon", "bottom shelf")
```

## 9.3 Baseline Prompts

### 9.3.1 `Cand+LLM-Q/A` Prompts

`Cand+LLM-Q/A` use the same prompts as `APRICOT` to generate candidate preferences (Sec 9.1.1) and plans (Sec 9.2). Below is the prompt that it used to determine whether to terminate, what is the best

question to ask if not terminating, and what is the best preference out of the candidate preference list if terminating. We use gpt-4o.

```
--------- System Message --------
You are a helpful, thoughtful assistant whose task is to select the user's
↪ preference from the demonstrations that the user provides and asking
↪ users for yes/no clarification question.
--------- Instruction --------
 # Input
 You are giving a list of preferences, a new initial condition of the
 ↪ fridge that your generated preference will get used at, and chat log
 ↪ of what you have asked so far.

 ## Preferences
 You are given a list of preferences that show how the user wants their
 ↪ fridge organized.
 - A preference is a short paragraph that specifies requirements for each
 ↪ category of grocery items.
 - There will be at least one requirement for each category. The type of
 ↪ requirement for each category can be different.

 ## New initial state of the fridge
 This is the new initial state that your preference will get used to
 ↪ determine how a new set of objects will get placed into the fridge and
 ↪  satisfy the user's preference.

 ## Objects to put away
 This is the list of object that will get put away in this new initial
 ↪ state of the fridge.

 ## Chat log
 This is a json that contains a history of your thought and your
 ↪ conversation with the user.
 - When the "role" is "thought", this is your internal reasoning about
 ↪ what you were going to do. The user never sees your thought.
 - When the "role" is "assistant", this is the yes/no clarification
 ↪ question that you ask the user about.
 - When the "role" is "user", this is the yes/no answer that the user
 ↪ provided to your yes/no clarification question.

 ## Can you still ask question?
 There is a maximum number of questions that you can ask. This field will
 ↪ tell you if you can still ask questions or if you must terminate and
 ↪ generate your best guess at the user's preference.

 # Goal
 Your goal is to select the best of the given preference in how the
 ↪ grocery items should be placed into the fridge. You can do this
 ↪ inference by analyzing the given preferences and asking users yes/no
 ↪ clarification questions. Doing this process, you can choose betwwen 2
 ↪ actions:
   (1) Ask the user a yes/no clarification question
   (2) Select the index of the best preference from the list and terminate

 # Instructions and useful information
 ## Set up of the fridge
 The fridge can be segmented into 3 shelves (top shelf, middle shelf,
 ↪ bottom shelf); each shelf has 2 sides (left side, right side):
```

```
- left side of the top shelf, right side of the top shelf
- left side of the middle shelf, right side of the middle shelf
- left side of the bottom shelf, right side of the bottom shelf

## Concrete steps that you must follow
To successfully determine the user's preference, you must do the
↪ following steps.
### Step 1: Reasoning about what actions to do
You can choose between 2 actions:
(1) Ask the user a yes/no clarification question
(2) Select the best preference, output its index, and terminate

You must follow these rules when you are deciding what to do:
- If you are still uncertain about the preference, you should ask an
↪ informative yes/no clarification question.
- If you are 100% confident about the preference, you can select which
↪ you think the right preference is and terminate.

### Step 2: Take the action that you decided to do
You must reply in a valid json format:
```json
{
  "terminate? (yes/no)": "<you must only reply yes or no>",
  "reasoning": "<you must write down the reasoning behind either why you
  ↪ generated a specific question (if you choose the first action) or why
  ↪  you think the user's preference is what you are generating (if you
  ↪ choose the second action)>",
  "question": "<you must only fill out this string if you chose action (1)
  ↪  ask yes/no clarification question. Otherwise, you must leave this as
  ↪  an empty string.>",
  "index_of_best_preference": <you must only fill out this if you chose
  ↪ action (2) select index of best preference. Otherwise, you must leave
  ↪  this as null.>
}
```

#### If you chose (1) Ask the user a yes/no clarification question in
↪ Step 3
You must follow these rules when coming up with questions to ask:
- You must ask simple yes or no questions.
- Your questions cannot ask if the user prefers A or B. However, you can
↪ ask if the user prefers A rather than B.
- Your questions must not be about overall objects or items. You must
↪ never ask questions like "Do you prefer to items to be ...?" or "Do
↪ you prefer items of the same category ...?"
- Your question must be about specifc categories or type of objects. The
↪ categories are for example: ["fruits", "vegetables", "dairy product",
↪ "condiments", "juice and drinks"]
- Your output must have "index_of_best_preference": null

Remember that you must reply in a valid json format:
```json
{
  "terminate? (yes/no)": "no",
  "reasoning": "<you must write down the reasoning behind why you
  ↪ generated a specific question (because you choose the first action)
  ↪ >",
  "question": "<you must fill out this string because you chose action (1)
  ↪  ask yes/no clarification question.>",
```

```
    "index_of_best_preference": null
}
```

#### If you chose (2) Select the best preference index
You must follow these rules.
- You must now terminate so you should answer "yes" under the key "
↪ terminate? (yes/no)"
- You must output the index of the best preference, with 0 referring to
↪ the first listed preference

Remember that you must reply in a valid json format:
```json
{
  "terminate? (yes/no)": "yes",
  "reasoning": "<you must write down the reasoning behind why you think
  ↪ the user's preference is what you are selecting (because you choose
  ↪ the second action)>",
  "question": "",
  "index_of_best_preference": <you must output a 0-indexed integer
  ↪ corresponding to the preference you picked from the given preference
  ↪ list because you chose action (2) select best preference.>
}
```
```

### 9.3.2  `LLM-Q/A` Prompts

`LLM-Q/A` only uses the prompt below to determine whether to terminate, determine what is the best question to ask if not terminating, and generate the preference based on demonstrations and its queries with the user if terminating. We use `gpt-4o`.

```
--------- System Message --------
You are a helpful, thoughtful assistant whose task is to select the user's
↪ preference from the demonstrations that the user provides and asking
↪ users for yes/no clarification question.
--------- Instruction --------
 # Input
 You are giving a list of preferences, a new initial condition of the
 ↪ fridge that your generated preference will get used at, and chat log
 ↪ of what you have asked so far.

 ## Preferences
 You are given a list of preferences that show how the user wants their
 ↪ fridge organized.
 - A preference is a short paragraph that specifies requirements for each
 ↪ category of grocery items.
 - There will be at least one requirement for each category. The type of
 ↪ requirement for each category can be different.

 ## New initial state of the fridge
 This is the new initial state that your preference will get used to
 ↪ determine how a new set of objects will get placed into the fridge and
 ↪  satisfy the user's preference.

 ## Objects to put away
 This is the list of object that will get put away in this new initial
 ↪ state of the fridge.
```

## Chat log
This is a json that contains a history of your thought and your
↪ conversation with the user.
- When the "role" is "thought", this is your internal reasoning about
↪ what you were going to do. The user never sees your thought.
- When the "role" is "assistant", this is the yes/no clarification
↪ question that you ask the user about.
- When the "role" is "user", this is the yes/no answer that the user
↪ provided to your yes/no clarification question.

## Can you still ask question?
There is a maximum number of questions that you can ask. This field will
↪ tell you if you can still ask questions or if you must terminate and
↪ generate your best guess at the user's preference.

# Goal
Your goal is to select the best of the given preference in how the
↪ grocery items should be placed into the fridge. You can do this
↪ inference by analyzing the given preferences and asking users yes/no
↪ clarification questions. Doing this process, you can choose betwwen 2
↪ actions:
  (1) Ask the user a yes/no clarification question
  (2) Select the index of the best preference from the list and terminate

# Instructions and useful information
## Set up of the fridge
The fridge can be segmented into 3 shelves (top shelf, middle shelf,
↪ bottom shelf); each shelf has 2 sides (left side, right side):
- left side of the top shelf, right side of the top shelf
- left side of the middle shelf, right side of the middle shelf
- left side of the bottom shelf, right side of the bottom shelf

## Concrete steps that you must follow
To successfully determine the user's preference, you must do the
↪ following steps.
### Step 1: Reasoning about what actions to do
You can choose between 2 actions:
(1) Ask the user a yes/no clarification question
(2) Select the best preference, output its index, and terminate

You must follow these rules when you are deciding what to do:
- If you are still uncertain about the preference, you should ask an
↪ informative yes/no clarification question.
- If you are 100% confident about the preference, you can select which
↪ you think the right preference is and terminate.

### Step 2: Take the action that you decided to do
You must reply in a valid json format:
```json
{
  "terminate? (yes/no)": "<you must only reply yes or no>",
  "reasoning": "<you must write down the reasoning behind either why you
  ↪ generated a specific question (if you choose the first action) or why
  ↪  you think the user's preference is what you are generating (if you
  ↪ choose the second action)>",
  "question": "<you must only fill out this string if you chose action (1)
  ↪  ask yes/no clarification question. Otherwise, you must leave this as
  ↪  an empty string.>",
```

```
  "index_of_best_preference": <you must only fill out this if you chose
  ↪ action (2) select index of best preference. Otherwise, you must leave
  ↪  this as null.>
}
```

#### If you chose (1) Ask the user a yes/no clarification question in
↪ Step 3
You must follow these rules when coming up with questions to ask:
- You must ask simple yes or no questions.
- Your questions cannot ask if the user prefers A or B. However, you can
↪ ask if the user prefers A rather than B.
- Your questions must not be about overall objects or items. You must
↪ never ask questions like "Do you prefer to items to be ...?" or "Do
↪ you prefer items of the same category ...?"
- Your question must be about specifc categories or type of objects. The
↪ categories are for example: ["fruits", "vegetables", "dairy product",
↪ "condiments", "juice and drinks"]
- Your output must have "index_of_best_preference": null

Remember that you must reply in a valid json format:
```json
{
  "terminate? (yes/no)": "no",
  "reasoning": "<you must write down the reasoning behind why you
  ↪ generated a specific question (because you choose the first action)
  ↪ >",
  "question": "<you must fill out this string because you chose action (1)
  ↪  ask yes/no clarification question.>",
  "index_of_best_preference": null
}
```

#### If you chose (2) Select the best preference index
You must follow these rules.
- You must now terminate so you should answer "yes" under the key "
↪ terminate? (yes/no)"
- You must output the index of the best preference, with 0 referring to
↪ the first listed preference

Remember that you must reply in a valid json format:
```json
{
  "terminate? (yes/no)": "yes",
  "reasoning": "<you must write down the reasoning behind why you think
  ↪ the user's preference is what you are selecting (because you choose
  ↪ the second action)>",
  "question": "",
  "index_of_best_preference": <you must output a 0-indexed integer
  ↪ corresponding to the preference you picked from the given preference
  ↪ list because you chose action (2) select best preference.>
}
```

### 9.3.3 `Non-Interactive` **Prompts**

`Non-Interactive` uses the prompt below to generate one preference from demonstrations, and it uses the same prompt as `APRICOT` to generate the task plan. We use `gpt-4` because this is a complex task to reason about all the demonstrations.

```
--------- System Message --------
You are an assistant who sees someone demonstrating how the fridge is
↪ organized and summarizes that person's preference.
--------- Instruction --------
 # Input
 You are given 2 demonstrations that show the before and after when a set
 ↪ of objects gets put into the fridge. For each demonstration:
 - "Objects that got put away" describes the objects that the user will
 ↪ demonstrate how they would like to put in the fridge.
 - "Initial state of the fridge" describes the objects that are initially
 ↪ in the fridge before the user starts the demonstration.
 - "Final state of the fridge" describes what the fridge looks like after
 ↪ the demonstration. All the objects in "Objects that got put away"
 ↪ should be in the fridge now.

 # Goal
 Your goal is to generate one preferences that are consistent with the
 ↪ demonstrations and explain what the user want.

 # Instructions and useful information
 ## Specific locations in the fridge
 The fridge can be segmented into 3 shelves (top shelf, middle shelf,
 ↪ bottom shelf); each shelf has 2 sides (left side, right side). The
 ↪ fridge has 6 specific locations:
 - left side of the top shelf, right side of the top shelf
 - left side of the middle shelf, right side of the middle shelf
 - left side of the bottom shelf, right side of the bottom shelf

 ## Details about the preferences that you need to output
 A preference is a short paragraph that specifies requirements for each
 ↪ category of grocery items. There must be at least one requirement for
 ↪ each category. The type of requirement for each category can be
 ↪ different. The categories are: "Fruits", "Vegetables", "Juice-and-soft
 ↪ -drinks", "Dairy-Products", and "Condiments".

 The requirement needs to be one of the following:
 - **Type-1. Specific Locations.** These represent that the object must
 ↪ place at this specific location. The options are:
   - "left side of top shelf"
   - "right side of top shelf"
   - "left side of middle shelf"
   - "right side of middle shelf"
   - "left side of bottom shelf"
   - "right side of bottom shelf".
 - **Type-2. General Locations.** These are vaguer locations that contain
 ↪ multiple specific locations. The options are:
   - "left side of fridge"
   - "right side of fridge"
   - "top shelf"
   - "middle shelf"
   - "bottom shelf"
 - **Type-3. Relative Positions.** The options are:
   - "<category> must be placed together next to existing <category>
   ↪ regardless of which shelf they are on."
```

```
  - "<category> must be placed on the same shelf next to <another category
  ↪  of objects>, and which specific shelf does not matter."

In addition to giving specific requirements for each category of grocery
↪ items, sometimes you may choose to add additional requirements. The
↪ options are:
- **Type-4. Exception For Attribute**
  - "<category> needs to be placed at <specific location 1>, but <
  ↪ attribute of category> needs to be placed at <specific location 2>."
  ↪ An attribute includes a subcategory of the object, the size/weight of
  ↪  the object, a specific feature of the object, etc.
- **Type-5. Conditional On Space**
  - "If there are less than <N> objects at <priminary specific location>,
  ↪ I want <category> to be placed at <priminary specific location>. Else
  ↪ , I want <category> to be placed at <second choice specific location
  ↪ >."

Now, you need to generate 1 preference as a jsons.
- Each preference must be in natural language.
- Each preference must contain at least one placement requirement for
↪ each category of objects: "fruits", "vegetables", "drinks", "dairy
↪ product", and "condiments".

You must refer to the demonstrations and output the preferences in this
↪ format
```json
{
  "reasoning": "<You must explain your thought process behind generating
  ↪ this preference.>",
  "preference": "<You must write the preference in natural language here>"
}
```
```

