# OpenReview forum: "APRICOT: Active Preference Learning and Constraint-Aware Task Planning with LLMs"
_robot-learning.org/CoRL/2024/Conference — CoRL 2024_

### Official Review · Reviewer_YWXV · 2024-07-15
**Review of "APRICOT: Active Preference Learning with Constraint-Aware Task Planner for Home Robots"**

**Originality:** 3
**Technical Quality:** 2
**Clarity Of Presentation:** 3
**Potential Impact:** 3
**Recommendation:** 3
**Confidence:** 4

**Review:**

The integration of LLM-based Bayesian active preference learning with constraint-aware task planning is innovative and addresses significant challenges in the field. The methodology is described in detail, providing a clear understanding of the approach and its components. The paper is generally clear, with a logical flow and detailed descriptions of the proposed approach. The integration of user preferences with environmental constraints through a novel three-stage approach is a notable contribution to the field. The significance of this work is substantial, as it addresses critical challenges in the deployment of home robots for personalized tasks. By enabling robots to better understand and adapt to user preferences and environmental constraints, the approach has the potential to improve the usability and effectiveness of home robots in real-world applications. The practical implications of this work are considerable, particularly in the context of assistive technologies and personalized robotics.

Weaknesses:

• While the paper discusses the ambiguity in interpreting user actions, the strategies for resolving such ambiguities could be elaborated further. Examples or case studies demonstrating how these ambiguities are handled would be useful.

• There is limited discussion on the scalability of the system with increasing complexity or the number of user demonstrations.

• The computational requirements of APRICOT in terms of processing time and memory usage are not thoroughly discussed. Understanding the system's computational efficiency is important for assessing its real-time performance capabilities.

**Quality Of The Limitations Section:**

2

**Questions For Rebuttal:**

• Could you provide more details on the integration process of APRICOT with real-robot systems? Are there specific hardware or software requirements?

• The paper mentions the challenge of inherent ambiguity in interpreting user actions. How does APRICOT specifically address this ambiguity?

**Robotics Focus:**

3

**Summary Of Paper:**

The paper presents APRICOT, a three-stage approach designed to help home robots perform personalized tasks by balancing user preferences and environmental constraints. APRICOT learns user preferences from visual demonstrations, generates plans that satisfy these preferences, and executes the plans using a real-robot system. The approach leverages LLM-based Bayesian active preference learning to handle ambiguities in user actions and dynamically adapts to environmental constraints. The effectiveness of APRICOT is demonstrated through evaluations on a diverse set of organization tasks and real-world scenarios.

**Summary Of Recommendation:**

The paper presents an innovative integration of LLM-based Bayesian active preference learning with constraint-aware task planning, addressing significant challenges in home robot task planning. The methodology is detailed and generally clear, providing a robust understanding of the approach. However, the paper could benefit from further elaboration on strategies for resolving ambiguities in user actions, discussions on scalability with increased complexity, and detailed computational requirements for assessing real-time performance. Despite these weaknesses, the contributions and potential impact of the work are substantial, making it a valuable addition to the field.

---

### Official Review · Reviewer_SyaR · 2024-07-20
**Novel method for learning user preferences using demonstration summaries and active clarifications.**

**Originality:** 4
**Technical Quality:** 4
**Clarity Of Presentation:** 5
**Potential Impact:** 3
**Recommendation:** 3
**Confidence:** 4

**Review:**

**Strengths**
- APRICOT is a principled solution for using LLM Planners to balance between preferences and constraints.
- The results help identify the role of different elements. The main results delineate the role of asking questions, having candidate preferences, and using information-gain based metric for clarifications. The preference-feasibility tradeoff study provides insight into how it makes this tradeoff.
- Quantifying ambiguity in the context of user preferences and their application to a task as well as use in active information gathering is crucial in real-world human-robot interaction, and doing so over LLM-generated preference representation is a novel contribution of this work.
- The paper is well-written in terms of clarity of the proposed method and motivation.

**Concerns**
- The paper mentions almost no information about how the datasets are generated, for  both the simulated setting as well as the real world setting. It is crucial to understand the results in context of whether the LLM-based method is evaluated on a realistic crowd-sourced dataset or one generated from the same LLM.
- Related to the above point, LLMs are used to both simulate a human preference and response in evaluations and to model expected information gain to ask the right question. Assuming the same LLM is used for both of these, is it fair to say that the model has access to a perfect model of the human, which it can query through its internal optimization loop? If so, why is the performance is not closer to 100%. What are failure cases of the model?
- I understand that it is difficult to find a matching baseline for such a problem, but the baselines used in the evaluations are effectively ablations of the method, (obtained by removing the information-gain clarification, candidate preferences, and the ability to ask questions). This leaves many questions in establishing the capability of the proposed method in context of prior work. Since the preferences are constrained to be a combination of one of the 5 types of atomic preferences (as outlined in the LLM prompts), is it not, in theory, possible to use predicate-combinations as preferences, and benchmark against a classical AI method composed of information-gain based clarification and classical AI planner? While this baseline might do worse, lacking the semantic knowledge of LLMs, a numerical comparison would certainly help understand the evaluations better.
- Can the authors include insight about what each iteration of asking clarification questions entails in terms of computational overhead, particularly in the plan rollouts needed to identify whether to ask a question.

**Edit after rebuttal**: The authors clarified necessary details, crucially providing an analysis of failure cases. Unfortunately, the absolute performance is not particularly high despite that the same LLM is used as the human evaluator and the model, and the performance relative to ablation-like baselines does not help. However, I understand that there are no other clear baselines that can be used, and the error breakdowns help make the evaluations more comprehensive. My evaluation continues to lean towards an acceptance.

**Quality Of The Limitations Section:**

3

**Questions For Rebuttal:**

- Can the authors elaborate on how the dataset is generated. Is it crowdsourced? Artificially generated using AI?
- How does the proposed method compares against classical AI methods, and alternatively, why is such a comparison not possible?
- What are the failure modes of the model, and, intuitively, why can the performance not be near perfect, since the model seems to have access to effectively a perfect human model. More generally, can the authors comment on the second point in the 'concerns' section above?

**Robotics Focus:**

4

**Summary Of Paper:**

This work proposes APRICOT to recover user preferences from demonstrations and active clarification, and utilize inferred preferences in a task while considering environmental constraints. The authors highlight two challenges with preference learning: the inherent ambiguity in inferring preferences from demonstrations, and potentially infeasible preferences. They approach this by reasoning over a closed loop with simulated human feedback, obtained from an LLM, as well as simulated environment feedback, obtained from a planner. The proposed method first employs a VLM to parse visual demonstrations into a textual representation, and uses an LLM to generate a set of candidate preferences based on the textual demonstrations. To disambiguate between different preference hypotheses, APRICOT generates a set of questions using an LLM, then asks the most informative question till the uncertainty collapses. Finally it executes the single remaining optimal plan.

**Summary Of Recommendation:**

The paper identifies core challenges in preference learning, proposes a formulation to address them, and does a great job at clearly detailing the proposed method. However, the evaluation section, particularly the dataset used, requires more details.

---

### Official Review · Reviewer_gKEt · 2024-07-21
**Interesting paper but many concerns**

**Originality:** 3
**Technical Quality:** 3
**Clarity Of Presentation:** 3
**Potential Impact:** 2
**Recommendation:** 3
**Confidence:** 4

**Review:**

The paper introduces a three-stage approach called APRICOT for solving personalized organizational tasks with environmental constraints. Unlike previous methods that use linear combinations of features, APRICOT represents preferences in natural language, which can capture complex preferences more effectively. The approach includes an interactive mechanism to refine preferences based on user feedback, addressing a gap in previous studies that lacked this feature. APRICOT bridges two paradigms by proposing candidate questions through LLMs and selecting the best questions to maximize information gain, unlike previous methods that rely solely on LLMs or traditional experimental designs. While the paper introduces some interesting ideas and makes incremental improvements, but leaving few concerns:
1/The performance improvements over baseline methods are relatively modest. For instance, achieving 58% accuracy in preference learning, while higher than baselines, is not a substantial leap. This raises questions about whether the approach represents a significant advancement in the field.
2/While the combination of natural language processing with active learning and task planning is interesting, the individual techniques themselves are not novel. The integration of these components does not introduce a groundbreaking new technique or paradigm that significantly advances the state of the art.
3/The model makes several assumptions, such as having a perfect world model and binary geometric constraints, which are simplifications of real-world scenarios. These assumptions reduce the practical novelty and applicability of the approach in more complex and realistic environments.

I also have some concerns in the methodology:
1/In the Objective Function (eq. 1):  the author assumes the availability of a perfect world model to determine the next state after an action is taken. This assumption is often unrealistic in practical scenarios, where the world model might be imperfect or incomplete. This could lead to incorrect state transitions and suboptimal plans.
2/In the Objective Function, the constraint  is binary, indicating whether geometric constraints are satisfied or not. However, in real-world scenarios, geometric constraints may not be binary and could require more nuanced handling (e.g., soft constraints or penalties for near-collisions).
3/The Disadvantage Function (eq. 2):  the maximization over xi assumes that the optimal plan for a given preference can be easily found, which may not always be the case. This could lead to computational complexity and intractability, especially for large action spaces. Need explanation of it.
4/The function quantifies the disadvantage of a plan compared to others in the library. However, it does not account for the variability and uncertainty in preference satisfaction, which can vary significantly across different user preferences and scenarios.
5/The Termination Condition, the choice of threshold is critical. How this threshold is got selected? In my point of view, if set too low, the algorithm might never terminate; if set too high, it might terminate prematurely with suboptimal plans. The paper does not provide a robust method for setting or adapting this threshold.

Empirically, in order to proof effectiveness of this model, the author shows some solid evidences where they showed their model achieves the highest preference accuracy of 58% compared to Non-Interactive (35%), LLM-Q/A (39%), and Cand+LLM-Q/A (43%) and it requires 71.9% fewer queries than LLM-Q/A and 46.25% fewer queries than Cand+LLM-Q/A. However, I have some concerns.
1/The baseline comparisons indicate that all models, including APRICOT, struggle with complex preferences. This raises questions about the practical utility of the approach.
2/The trade-off between query efficiency and preference accuracy needs more exploration.
3/Additionally, the significant reduction in queries could indicate that APRICOT may not be capturing sufficient information to make accurate inferences.

Other suggestions:
1/Several sentences are long and complex. Breaking them into simpler sentences can improve readability.
2/ there is frequent use of passive voice. Where possible, use active voice to make sentences clearer. For example, on page 3, line 80, “is represented” can be changed to “represents”.
3/ The mathematical notation should be reviewed for consistency. For example, on page 3, lines 89-90, ensure that all symbols are defined when first introduced.
4/ All standalone equations should end with proper punctuation signs.
5/ In Eq. (4), the suffix and prefix of summation are inaccurately used. They should be sum_{o\in{yes, no}} and sum_{i=1}^N.

**Quality Of The Limitations Section:**

2

**Questions For Rebuttal:**

see in the review.

**Robotics Focus:**

2

**Summary Of Paper:**

The paper introduces a three-stage approach called APRICOT for solving personalized organizational tasks with environmental constraints. There are many concerns in the methodology part and some issues in experiments,.

**Summary Of Recommendation:**

see in the review

---

### Author Rebuttal · Authors · 2024-08-07

We thank the reviewers for their suggestions on how to improve our paper. We have attached the revised draft and the performance-bound proof below.

---

### Decision · Program_Chairs · 2024-09-04

**Decision:**

Accept

**Comment:**

**Summary**

The paper introduces APRICOT, a three-stage framework for enabling home robots to perform personalized tasks while accounting for user preferences and environmental constraints. APRICOT addresses two key challenges: the ambiguity in inferring preferences from user demonstrations and the risk of generating infeasible preferences. The method begins by using a VLM to convert visual demonstrations into text, followed by an LLM that generates candidate preferences. To resolve ambiguities, APRICOT actively asks targeted questions generated by the LLM until the most accurate preference is identified. Finally, it executes the optimal plan that satisfies the clarified preferences. Despite some initial concerns with the methodology and experiments, APRICOT demonstrates effectiveness in various organization tasks through evaluations on real-world robotic systems.

**Review Summary**

*Strengths*:
- The problem of quantifying ambiguity of user preferences is important and relevant to the CoRL community.
- Reviewers thought the approach was interesting, principled, and well described.
- Some of the reviewers found the method novel and a substantial contribution.
- The experiments test the role of each component of the pipeline.
- The paper is well-written in terms of clarity of the method and motivation.

*Weaknesses*:
- Unclear applicability to real-world robotics. Some of the reviewers raised concerns regarding the applicability of the method to the real world. One reviewer made a point about potentially unrealistic assumptions (perfect world model, binary geometric constraints, optical plan can be found, etc.) that over-simplify the problem setting, to the point of reducing the practical novelty and applicability of the approach in more complex and realistic environments. Another reviewer raises issues about the system’s scalability and computational requirements.
- Contested novelty. While some of the reviewers found the integration of the individual components to be novel and significant, others raised the point that the components themselves are not new and simply putting them together is not particularly groundbreaking. The authors should comment in their rebuttal on why integrating these existing components together could be seen as novel and substantive.
- Insufficient baselines. One reviewer pointed out that the baselines used are effectively ablations of the method. Reviewers asked for a comparison with classical AI methods or alternatively a justification for why such a comparison would not be possible.
- Insufficient discussion. Reviewers asked for more discussion of the experiments and results that the authors should address: how is the data generated, what are the model’s failure modes and why does it struggle with complex preferences, what are strategies for resolving ambiguity, etc. Another point was raised about why the model is not performing better, especially since it appears that the method effectively has access to a perfect human model.
- (minor) Writing could be improved: reviewers have some suggestions for how to improve the grammar, sentence structure, and use of active voice in the paper.

**Outcome and Post-rebuttal**

After many efforts from the authors to address the reviewer comments, from failure case discussion to clearer contributions to applicability to robotics, it seems like all reviewers are in agreement about accepting this paper. Conditioned on including all the changes discussed in the rebuttal phase, I am in favor of accepting this paper.